# POMC neurons control fertility through differential signaling of MC4R in kisspeptin neurons

Rajae Talbi[1,2†], Todd L Stincic[3†‡], Kaitlin Ferrari[2], Choi Ji Hae[2†], Karol Walec[2], Elizabeth Medve[2], Achi Gerutshang[2], Silvia Leon[1,2], Elizabeth A McCarthy[1,2], Oline K Rønnekleiv[3,4], Martin J Kelly[3,4]*, Victor M Navarro[1,2,5]*

[1]Harvard Medical School, Boston, United States; [2]Division of Endocrinology, Diabetes and Hypertension, Department of Medicine, Brigham and Women's Hospital, Boston, United States; [3]Department of Chemical Physiology and Biochemistry, Oregon Health & Science University, Portland, United States; [4]Division of Neuroscience, Oregon National Primate Research Center, Beaverton, United States; [5]Harvard Program in Neuroscience, Boston, United States

*For correspondence:
kellym@ohsu.edu (MJK);
vnavarro@bwh.harvard.edu
(VMN)

[†]These authors contributed
equally to this work

Present address: [‡]Department
of Biology, Appalachian State
University, Boone, United States

Competing interest: The authors
declare that no competing
interests exist.

Reviewing Editor: Ashley Webb,
Buck Institute for Research on
Aging, United States

## eLife Assessment

The study presents **compelling** evidence that the melanocortin system originating in the arcuate nucleus of the hypothalamus plays a crucial role in puberty onset, representing a significant advance in our understanding of reproductive biology. The research employs innovative approaches and benefits from the combined expertise of two respected laboratories, enhancing the robustness of the findings. Given the potential impact on human health and the strength of the evidence presented, this **fundamental** work will likely influence the field substantially and may inform future clinical applications.

**Abstract** Inactivating mutations in the melanocortin 4 receptor (*MC4R*) gene cause monogenic obesity. Interestingly, female patients also display various degrees of reproductive disorders, in line with the subfertile phenotype of Mc4r KO female mice. However, the cellular mechanisms by which MC4R regulates reproduction are unknown. Kiss1 neurons directly stimulate gonadotropin-releasing hormone (GnRH) release through two distinct populations: the Kiss1[ARH] neurons, controlling GnRH pulses, and the sexually dimorphic Kiss1[AVPV/PeN] neurons controlling the preovulatory luteinizing hormone (LH) surge. Here, we show that *Mc4r* expressed in Kiss1 neurons regulates fertility in females. In vivo, deletion of *Mc4r* from Kiss1 neurons in female mice replicates the reproductive impairments of Mc4r KO mice without inducing obesity. Conversely, re-insertion of *Mc4r* in Kiss1 neurons of Mc4r null mice restores estrous cyclicity and LH pulsatility without reducing their obese phenotype. In vitro, we dissect the specific action of Mc4r on Kiss1[ARH] versus Kiss1[AVPV/PeN] neurons and show that Mc4r activation excites Kiss1[ARH] neurons through direct synaptic actions. In contrast, Kiss1[AVPV/PeN] neurons are normally inhibited by MC4R activation except under elevated estradiol levels, thus facilitating the activation of Kiss1[AVPV/PeN] neurons to induce the LH surge driving ovulation in females. Our findings demonstrate that POMC[ARH] neurons acting through MC4R directly regulate reproductive function in females by stimulating the 'pulse generator' activity of Kiss1[ARH] neurons and restricting the activation of Kiss1[AVPV/PeN] neurons to the time of the estradiol-dependent LH surge, and thus unveil a novel pathway of the metabolic regulation of fertility by the melanocortin system.

## Introduction

Obesity rates have skyrocketed in Western societies in the last decades, resulting in an alarming rise in comorbidities that place a significant burden on healthcare systems. The increase in obesity correlates with a decrease in fertility rates, leading to conception challenges that are experienced by approximately 15% of couples in the United States currently (*Health, 2020*).

The melanocortin 4 receptor (MC4R) binds (1) α-melanocyte stimulating hormone (αMSH), an agonist product of the pro-opiomelanocortin (*Pomc*) gene, and (2) the inverse agonist, agouti-related peptide (AgRP), to regulate food intake and energy expenditure (*Andermann and Lowell, 2017*; *Cone, 2006*). While the role of MC4R on food intake is largely mediated by neurons located in the paraventricular nucleus of the hypothalamus (PVN) (*Shah et al., 2014*), its expression in the brain is widespread (*Wang et al., 2020*) with the specific function of the different MC4R-expressing neurons in areas beyond the PVN remaining to be fully explored. Inactivating mutations in *MC4R* are a leading cause of monogenic obesity in humans (*Farooqi et al., 2003*) and cause excessive obesity and hyperphagia in mice (*Balthasar et al., 2005*). Scant evidence in humans shows an association between *MC4R* mutations and higher incidence of hypogonadotropic hypogonadism (*Hainerová et al., 2011*), alterations in the timing of puberty onset (*Doulla et al., 2014*), and polycystic ovary syndrome (PCOS; *Batarfi et al., 2019*); however, other studies have shown no association between MC4R and reproductive disorders (*Farooqi et al., 2003*). Therefore, the role of MC4R signaling in reproductive function in humans remains controversial despite the clear association found in mice. Supporting this role of MC4R, *Mc4r* null mice display an array of reproductive abnormalities that largely affects females, characterized by irregular estrous cycles, disrupted luteinizing hormone (LH) secretion, reduced corpora lutea, and reduced fertility (*Chen et al., 2017*; *Sandrock et al., 2009*; *Cui et al., 2022*). Further evidence from mice demonstrates that MC4R agonists robustly increase LH release in a kisspeptin-dependent manner (*Manfredi-Lozano et al., 2016*).

Kisspeptin is the most potent gonadotropin-releasing hormone (GnRH) secretagogue known to date, and it is mainly produced in two distinct neuronal populations. Kiss1 neurons of the arcuate nucleus of the hypothalamus (Kiss1[ARH]), present in both sexes, control the pulsatile (tonic) release of GnRH, and sex steroids attenuate their release of kisspeptin. Kiss1 neurons of the anteroventral periventricular continuum area (Kiss1[AVPV/PeN]) are predominantly present in females and are responsible for generating the preovulatory GnRH/LH surge essential for ovulation (*Goodman et al., 2022*). Despite these critical roles of both populations of Kiss1 neurons for reproduction, the underlying mechanisms that determine how each Kiss1 population responds differently to sex steroids to regulate the GnRH tonic versus surge release remain unresolved.

In rodents, compelling evidence indicates a close interaction between Kiss1 neurons and the melanocortin system: (1) fibers from POMC neurons in the arcuate nucleus (POMC[ARH]) project to and juxtapose Kiss1[ARH] neurons *Manfredi-Lozano et al., 2016*; (2) melanocortin signaling through MC4R contributes to the permissive role of leptin on puberty onset (*Manfredi-Lozano et al., 2016*; *Israel et al., 2012*; *Manfredi-Lozano et al., 2018*); and (3) Mc4r expression on both Kiss1[ARH] (*Cravo et al., 2011*; *Lam et al., 2021*; *Villa et al., 2024*) and Kiss1[AVPV/PeN] (*Cravo et al., 2011*; *Stephens and Kauffman, 2021*). Altogether, this evidence suggests a clear role for Mc4r in Kiss1 neurons, in the control of reproductive maturation and fertility.

In this study, we investigated the contribution of MC4R signaling versus obesity per se in the etiology of the reproductive impairments observed in *Mc4r* null mice, which could explain similar impairments observed in humans. Using genetic mouse models with specific deletion or re-insertion of *Mc4r* in Kiss1 neurons, we show that Mc4r action in Kiss1 neurons is necessary for normal reproductive function in female mice. Whole-cell voltage-clamp recordings evidenced an excitatory action of Mc4r on Kiss1[ARH] neurons and an estradiol-dependent inhibitory action on Kiss1[AVPV/PeN] neurons, with important implications for the timing of the preovulatory LH surge.

## Results

### *Mc4r* within Kiss1 neurons determines the timing of puberty onset in females

The expression of *Mc4r* within Kiss1[ARH] and Kiss1[AVPV/PeN] neurons has already been described elsewhere (*Cravo et al., 2011*; *Lam et al., 2021*; *Villa et al., 2024*; *Stephens and Kauffman, 2021*), suggesting

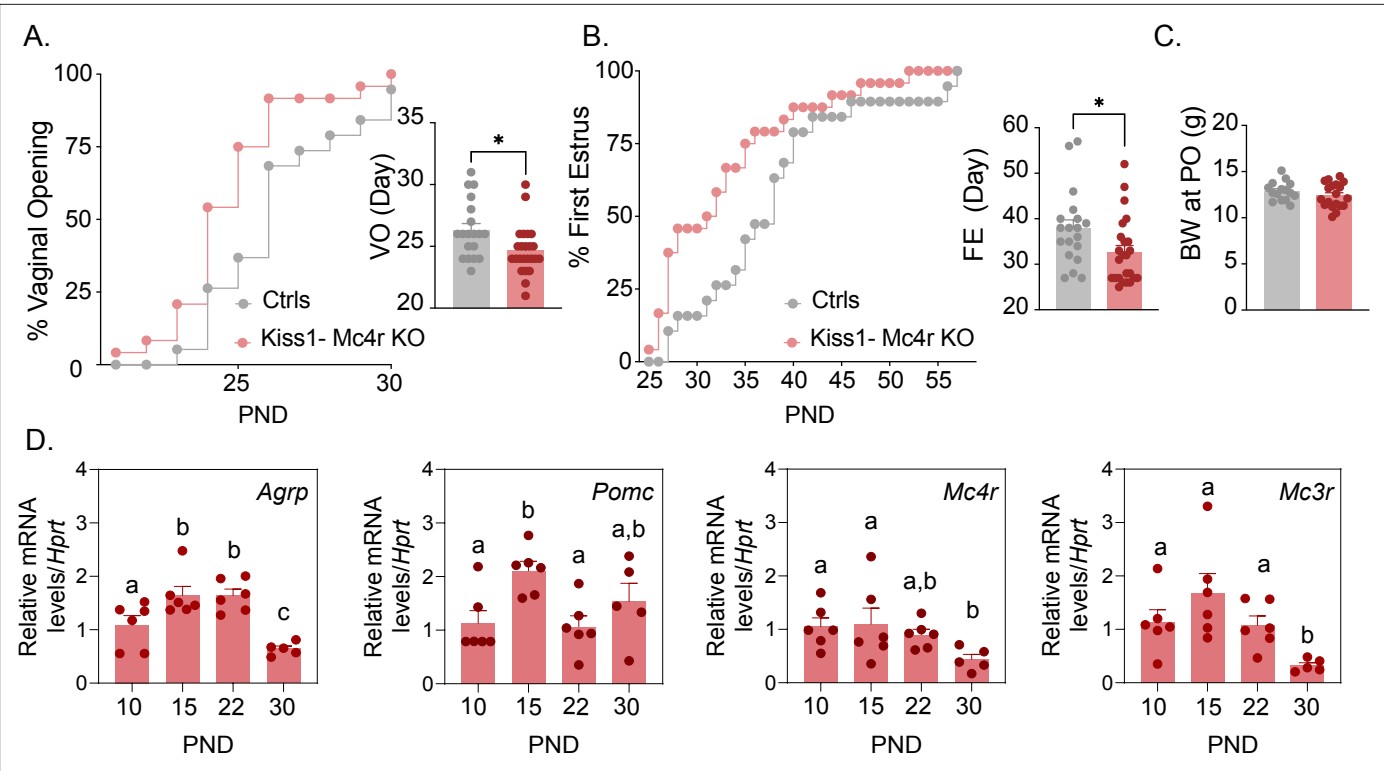

**Figure 1.** *Mc4r* expressed in Kiss1 neurons determines the timing of puberty onset. Kiss1- Mc4r KO females display advanced puberty onset, assessed by daily monitoring of vaginal opening (**A**) and first estrus (**B**), as documented by cumulative percent and mean age of animals at vaginal opening (**A**) and first estrus (**B**) in Kiss1- Mc4r KO females (n = 24) compared to WT littermates (n = 19). *p < 0.05 by Student's *t*-test. Data presented as the mean ± SEM. (**C**) Female Kiss1- Mc4r KO (n = 19) have normal body weight at the time of puberty onset compared to their WT littermates (n = 15). (**D**) Ontogeny expression of melanocortin genes (*Agrp*, *Pomc*, *Mc4r*, and *Mc3r*) in the ARH of WT female mice at different postnatal pre-pubertal and pubertal ages: P10, P15, P22, and P30, normalized to the housekeeping gene *Hprt* (n values: females at P10 (n = 6), P15 (n = 6), P22 (n = 6), and P30 (n = 5)). Groups with different letters are significantly different, as determined by one-way ANOVA followed by Fisher's LSD test. Data presented as the mean ± SEM.

The online version of this article includes the following figure supplement(s) for figure 1:

**Figure supplement 1.** Validation of the Kiss1- Mc4r KO mouse model.

a role for MC4R in the regulation of fertility. To further investigate the specific role of Mc4r in Kiss1 neurons, we generated a Kiss1-specific Mc4r knockout mouse model (Kiss1- Mc4r KO). The specific deletion of *Mc4r* from Kiss1 neurons, as well as the absence of global recombination, was confirmed through RNAscope. While Kiss1 neurons of Kiss1- Mc4r KO mice lack *Mc4r* transcript compared to their control littermates (*Figure 1—figure supplement 1A*), *Mc4r* was detectable in the PVN of all mice, which is a major site of *Mc4r* expression in the brain in the regulation of metabolism, therefore supporting the specific deletion of *Mc4r* only within Kiss1 neurons (*Figure 1—figure supplement 1B*).

Puberty onset was assessed daily from weaning age through the monitoring of vaginal opening (VO) and first estrus (FE). Kiss1- Mc4r KO females showed a significant advancement in the age of VO (p = 0.0150, Kiss1- Mc4r KO: 24.67 ± 0.39 vs. controls: 26.32 ± 0.53) and FE (p = 0.0341, Kiss1- Mc4r KO: 32.63 ± 1.4 vs. controls: 37.84 ± 1.90) compared to their littermate controls (*Figure 1A, B*). Body weight of Kiss1- Mc4r KO females at the age of puberty onset was similar between groups (p = 0.2596, Kiss1- Mc4r KO: 12.41 ± 0.30 vs. controls: 12.89 ± 0.26) (*Figure 1C*). These data suggest that melanocortin signaling on Kiss1 neurons participates in the timing of puberty onset in females. Since the role of the melanocortin system on puberty onset is largely unexplored, we evaluated the expression profile of the main components of this system in the hypothalami of WT female mice at post-natal days (PND) 10, 15, 22, and 30. Interestingly, expressions of *Agrp*, *Mc3r*, and *Mc4r* were significantly lower at the time of puberty onset (PND30) compared to earlier developmental ages (*Figure 1D*). This supports our in vivo data and suggests that a decrease in hypothalamic melanocortin signaling

drives puberty onset, in line with the advancement in the age of VO and FE observed in Kiss1- Mc4r KO female mice (*Figure 1A, B*).

## Mc4r in Kiss1 neurons is required for female reproduction

Because Kiss1- Mc4r KO females showed altered puberty onset, we further investigated their reproductive phenotype. Interestingly, Kiss1- Mc4r KO female mice displayed normal BW throughout the time of the study (up to PND150) (*Figure 2A*); however, they presented with irregular estrous cycles with predominantly more time spent in diestrus (p = 0.0023) and less time spent in estrus (p = 0.0053) than their control littermates (*Figure 2B*, *Figure 2—figure supplement 1A, B*). The assessment of the pulsatile release of LH every 10 min over 180 min revealed no change in the total number of pulses between groups but a significant increase in basal LH release (p = 0.036) in Kiss1- Mc4r KO female mice (0.42 ± 0.03) compared to controls (0.34 ± 0.02) (*Figure 2C–H*). The analysis of the gene expression of the 'KNDy' systems in the ARH, which control the GnRH pulse generator (*Goodman et al., 2022*), revealed normal expression levels of *Kiss1*, *Tac2*, and *Tacr3*, but significantly lower expression of *Pdyn* (p = 0.0067, Kiss1- Mc4r KO: 0.78 ± 0.038 vs. controls: 1.00 ± 0.019) (*Figure 2I–L*). The conserved expression of *Kiss1*, *Tac2*, and *Tacr3* correlates with the preserved LH pulse frequency and amplitude in Kiss1- Mc4r KO mice, while the lower inhibitory tone of dynorphin (*Figure 2I*) correlates with the higher basal LH levels (*Figure 2G*). However, the mRNA expression of these genes, alone, does not necessarily reflect changes in their activities. To assess the contribution of MC4R signaling in Kiss1 neurons to the induction of ovulation through the preovulatory LH surge, Kiss1- Mc4r KO females and control littermates were submitted to an LH surge induction protocol that showed the induced LH surge was significantly blunted in the Kiss1- Mc4r KO females (p = 0.0091), while the protocol clearly evoked the expected afternoon rise of LH in control mice (*Figure 2M*). In line with these findings, the ovaries of Kiss1- Mc4r KO females displayed fewer corpora lutea, markers of recent ovulation (p = 0.0054, Kiss1- Mc4r KO: 0.80 ± 0.37 vs. controls: 2.80 ± 0.37), in addition to increased cystic follicles (p = 0.0337, Kiss1- Mc4r KO: 1.60 ± 0.40 vs. controls: 0.40 ± 0.24) (*Figure 2N–P*), which correlate with decreased fertility as observed by the extended time to deliver pups (i.e., longer time to get pregnant) (p = 0.0030, Kiss1- Mc4r KO: 30.30 ± 4.60 vs. controls: 20.88 ± 0.22), and fewer pups per litter (p = 0.0095, Kiss1- Mc4r KO: 6.66 ± 0.55 vs. controls: 8.44 ± 0.24) (*Figure 2S, T*).

The increase in serum LH levels and decreased ovulation observed in the Kiss1- Mc4r KO females is reminiscent of PCOS mouse models (*Moore et al., 2013*; *McCarthy et al., 2022*). Thus, we investigated whether Kiss1- Mc4r KO females display a PCOS-like phenotype. We analyzed circulating levels of testosterone (T) and anti-Müllerian hormone (AMH), which are frequently elevated in PCOS models (*Dewailly et al., 2020*). The Kiss1- Mc4r KO females expressed normal T and AMH levels compared to control mice in diestrus (*Figure 2Q, R*). Thus, we can exclude a PCOS-like reproductive phenotype mediated by the lack of melanocortin signaling on Kiss1 neurons.

## Re-insertion of Mc4r in Kiss1 neurons of Mc4r KO mice improves reproductive function

Kiss1- Mc4r KO females displayed reproductive abnormalities resembling those described in Mc4r KO females (*Chen et al., 2017*; *Sandrock et al., 2009*; *Cui et al., 2022*). Thus, we hypothesized that the reproductive defects described for the Mc4r KO mice would be, at least in part, due to the absence of MC4R signaling in Kiss1 neurons. To further investigate this hypothesis, we generated mice that do not express *Mc4r* anywhere (*Mc4r^{loxTB}*, i.e., Mc4r KO) or that express *Mc4r* only in Kiss1 neurons (*Kiss1^{Cre}: Mc4r^{loxTB}*). The specific re-insertion of *Mc4r* within Kiss1 neurons in the *Kiss1^{Cre}: Mc4r^{loxTB}* mice was confirmed through RNAscope (*Figure 3—figure supplement 1A*). *Mc4r* expression was not detected in the PVN of these mice (*Figure 3—figure supplement 1B*), and it was only detected in Kiss1 neurons in the *Kiss1^{Cre}: Mc4r^{loxTB}* mice. Puberty onset was assessed daily from weaning age through the monitoring of VO and FE. *Mc4r^{loxTB}* and *Kiss1^{Cre}: Mc4r^{loxTB}* female mice displayed normal timing of puberty onset compared to their control littermates, as assessed by VO (p = 0.104, *Mc4r^{loxTB}*: 30.86 ± 1.90 vs. *Mc4r^{loxTB}*: 30.09 ± 1.00 vs. controls: 27.57 ± 0.81), and FE (p = 0.8472, *Mc4r^{loxTB}*: 35.14 ± 2.46 vs. *Kiss1^{Cre}: Mc4r^{loxTB}*: 34.55 ± 1.64 vs. controls: 35.80 ± 0.84) (*Figure 3A, B*), despite displaying significantly higher body weight at the time of puberty onset (p = 0.0001, *Mc4r^{loxTB}*: 15.49 ± 0.50 vs. *Kiss1^{Cre}: Mc4r^{loxTB}*: 15.54 ± 0.40 vs. Controls: 13.29 ± 0.30) (*Figure 3C*). As expected, *Mc4r^{loxTB}* females (Mc4r KO) displayed increased body weight (*Figure 3D*) and an array of reproductive impairments

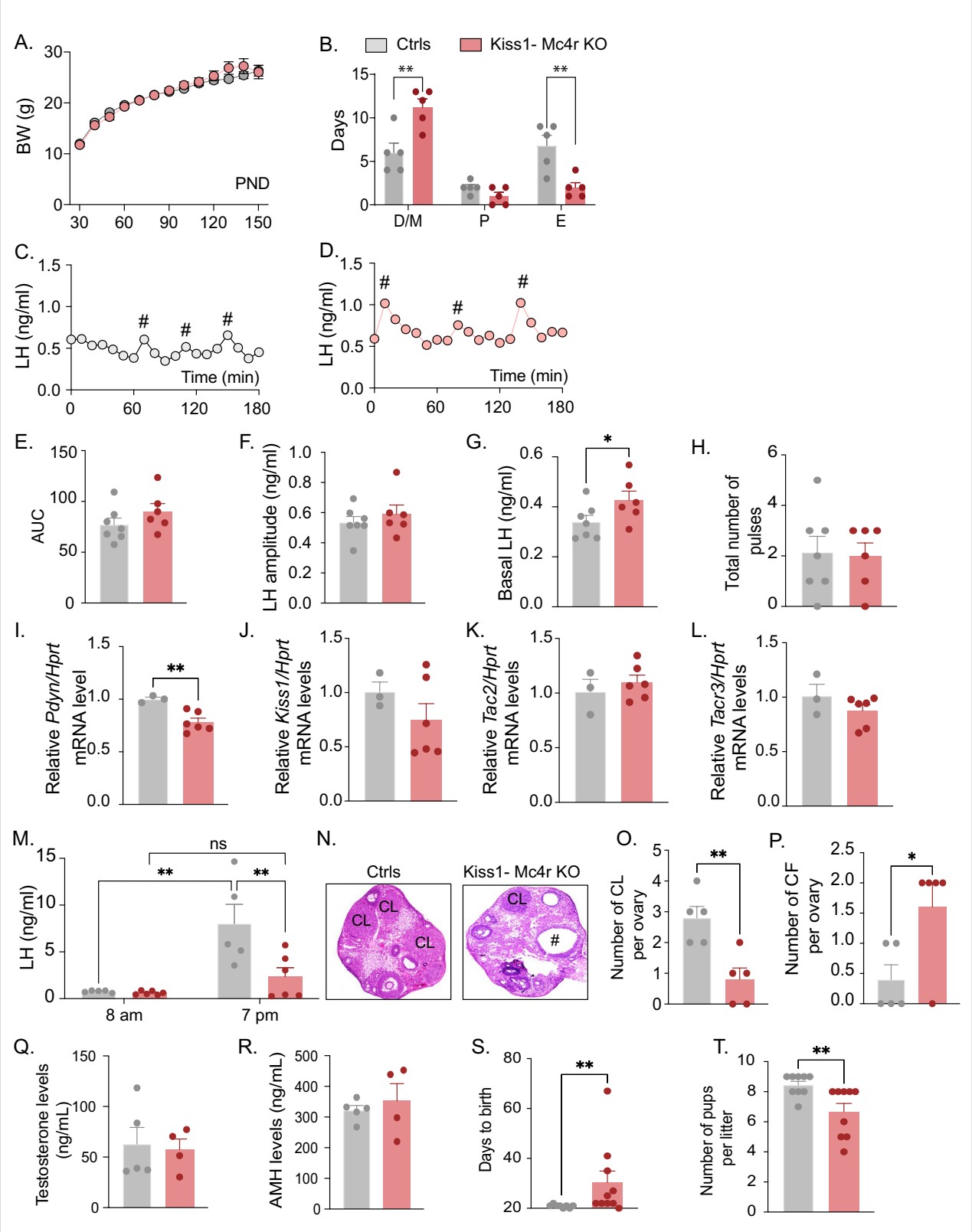

**Figure 2.** Deletion of Mc4r from Kiss1 neurons impairs fertility in Kiss1- Mc4r KO females. (**A**) Female Kiss1- Mc4r KO (*n* = 31) have normal body weight from weaning and until post-natal day (PND) 150 compared to their WT littermates (*n* = 26). (**B**) Kiss1- Mc4r KO females displayed irregular estrous cycles with a longer time in diestrus and a shorter time in estrus compared to control females. **p < 0.01, two-way ANOVA followed by Tukey's multiple comparisons test. (**C, D**) Pattern of luteinizing hormone (LH) pulsatility was analyzed in gonad intact Kiss1- Mc4r KO (*n* = 6) and control (*n* = 7) females.

*Figure 2 continued on next page*

*Figure 2 continued*

#Represents LH pulses. (**E**) LH total secretory mass assessed by area under the curve (AUC), (**F**) LH pulse amplitude, (**G**) basal LH, and (**H**) total number of pulses/180 min were analyzed. *p < 0.05 by Student's *t*-test. The expression of the KNDy genes *Pdyn* (**I**), *Kiss1* (**J**), *Tac2* (**K**), and *Tacr3* (**L**) was assessed in the ARH of adult Kiss1- Mc4r KO (n = 6) and control (n = 3) females. **p < 0.01 by Student's *t*-test. Data presented as the mean ± SEM. (**M**) Kiss1- Mc4r KO (n = 6) and control (n = 5) females were subjected to an LH surge induction protocol. LH samples were collected in the morning (AM [8 a.m.]) and evening (PM [7 p.m.]) after lights off. **p < 0.01, two-way ANOVA followed by Sidak multiple comparisons test. (**N**) Ovarian histology shows a decrease in the number of corpora lutea (CL), (**O**) and an increase in the number of cystic follicles (CF, **P**) of Kiss1- Mc4r KO compared to controls (n = 5/group). # represents cystic follicles. Serum levels of testosterone (**Q**) and anti-Müllerian hormone (AMH) (**R**) in adult gonad intact Kiss1- Mc4r KO and control females (n = 5/group). Student's *t*-test for unpaired samples. Data presented as the mean ± SEM. Kiss1- Mc4r KO females (n = 10) display impaired fertility compared to controls (n = 8) characterized by increased time to deliver pups (**S**) and decreased number of pups per litter (**T**), (n = 9/group). **p < 0.01 by Student's *t*-test. Data presented as the mean ± SEM.

The online version of this article includes the following figure supplement(s) for figure 2:

**Figure supplement 1.** Estrous cycles of the Kiss1- Mc4r KO, *Kiss1^Cre^: Mc4r^loxTB^*, and *Mc4r^loxTB^* mouse models.

that included: irregular estrous cycles with significantly more days in diestrus (p = 0.0016) and fewer days in estrus (p = 0.0317) compared to controls (*Figure 3E, Figure 2—figure supplement 1D, E*), significantly higher area under the curve (AUC) (*Mc4r^loxTB^*: 109.6 ± 2.39 vs. Controls: 80.36 ± 9.05) serum LH levels characterized by higher basal (*Mc4r^loxTB^*: 0.48 ± 0.02 vs. Controls: 0.37 ± 0.06) and amplitude levels per LH pulse (*Mc4r^loxTB^*: 0.730 ± 0.009 vs. Controls: 0.58 ± 0.07) (*Figure 3F–J*), and fewer corporal lutea (p = 0.0366, *Mc4r^loxTB^*: 0.80 ± 0.80 vs. Controls: 4.20 ± 1.15) (*Figure 3K, L*), recapitulating the same reproductive phenotype observed in Kiss1- Mc4r KO mice despite the differences in BW. Re-introduction of MC4R into Kiss1 neurons in *Kiss1^Cre^: Mc4r^loxTB^* mice completely recovered estrous cyclicity, which was similar to controls (diestrus: p = 0.1932, proestrus: p = 0.8262, estrus: p = 0.0547, compared to controls; *Figure 3E, Figure 2—figure supplement 1C, D*) despite *Kiss1^Cre^: Mc4r^loxTB^* mice showing the same degree of obesity as *Mc4r^loxTB^* mice (*Figure 3D*), indicating that obesity per se was not mediating the irregular estrous cycles in Mc4r KO mice. As indicated above, *Mc4r^loxTB^* mice display higher overall circulating LH levels, and this feature was also recovered by the re-insertion of Mc4r in Kiss1 neurons of *Kiss1^Cre^: Mc4r^loxTB^* mice (basal LH: 0.32 ± 0.04, LH amplitude: 0.48 ± 0.04, LH pulses/180 min: 2.40 ± 0.67 and AUC: 70.62 ± 7.15 compared to controls; *Figure 3G–J*), further confirming that the melanocortin action on Kiss1 neurons is required for the normal control of gonadotropin release. Despite these significant improvements in reproductive function, *Kiss1^Cre^: Mc4r^loxTB^* mice presented fewer corpora lutea than controls (0.80 ± 0.80), similar to *Mc4r^loxTB^* mice, suggesting that an ovulatory impairment persists (*Figure 3K, L*). While tonic LH release and estrous cyclicity are predominantly controlled by Kiss1^ARH^ neurons, ovulation is mediated by the induction of the preovulatory LH surge by Kiss1^AVPV/PeN^ neurons. It is possible that the metabolic signals derived from their obese phenotype, and/or the absence of the direct action of MC4R in GnRH neurons (*Israel et al., 2012*; *Roa and Herbison, 2012*) prevents the complete recovery of ovulation in *Kiss1^Cre^: Mc4r^loxTB^* mice. Both genetic models displayed significantly lower T (*Mc4r^loxTB^*: 28.50 ± 6.20 vs. *Kiss1^Cre^: Mc4r^loxTB^*: 31.22 ± 3.90 vs. Controls: 62.84 ± 16.53) and AMH (*Mc4r^loxTB^*: 150.6 ± 22.0 vs. *Mc4r^loxTB^*: 146.9 ± 10.8 vs. Controls: 321.8 ± 15.7) levels than control mice in diestrus (*Figure 3M, N*). Therefore, we can exclude, once again, a PCOS-like reproductive phenotype mediated by the lack of melanocortin signaling.

## Kiss1^ARH^ neurons are excited by Mc4r agonists

Based on the expression of *Mc4r* in Kiss1 neurons and the reproductive impairment found after *Mc4r* deletion within Kiss1 neurons (*Figure 2*), we hypothesized that Kiss1^ARH^ neurons are direct targets of melanocortins and would respond to MC4R activation. Initially, we did whole-cell, voltage-clamp recordings in Kiss1^ARH^ neurons (*Figure 4A*). *Kiss1^Cre^* x Ai32 or *Kiss1^Cre^* AAV-injected mice underwent ovariectomy (OVX) and were given estradiol (E2) replacement (*see Methods*). We targeted fluorescent cells for recording, which were isolated synaptically by bathing the slices in tetrodotoxin (TTX, 1 µM). Focal application of the high-affinity melanocortin receptor agonist melanotan II (MTII, ~250 nM) elicited a small inward current in half of the isolated Kiss1^ARH^ cells (10/21) (*Figure 4B*). Next, we examined if the E2 state affected MTII response by recording from Kiss1^ARH^ neurons from OVX females and found two-thirds (6/9 cells) responded, but there was no significant difference in the average current (*Figure 4C*). Finally, it must be noted that at this concentration, MTII is a non-selective MCR agonist,

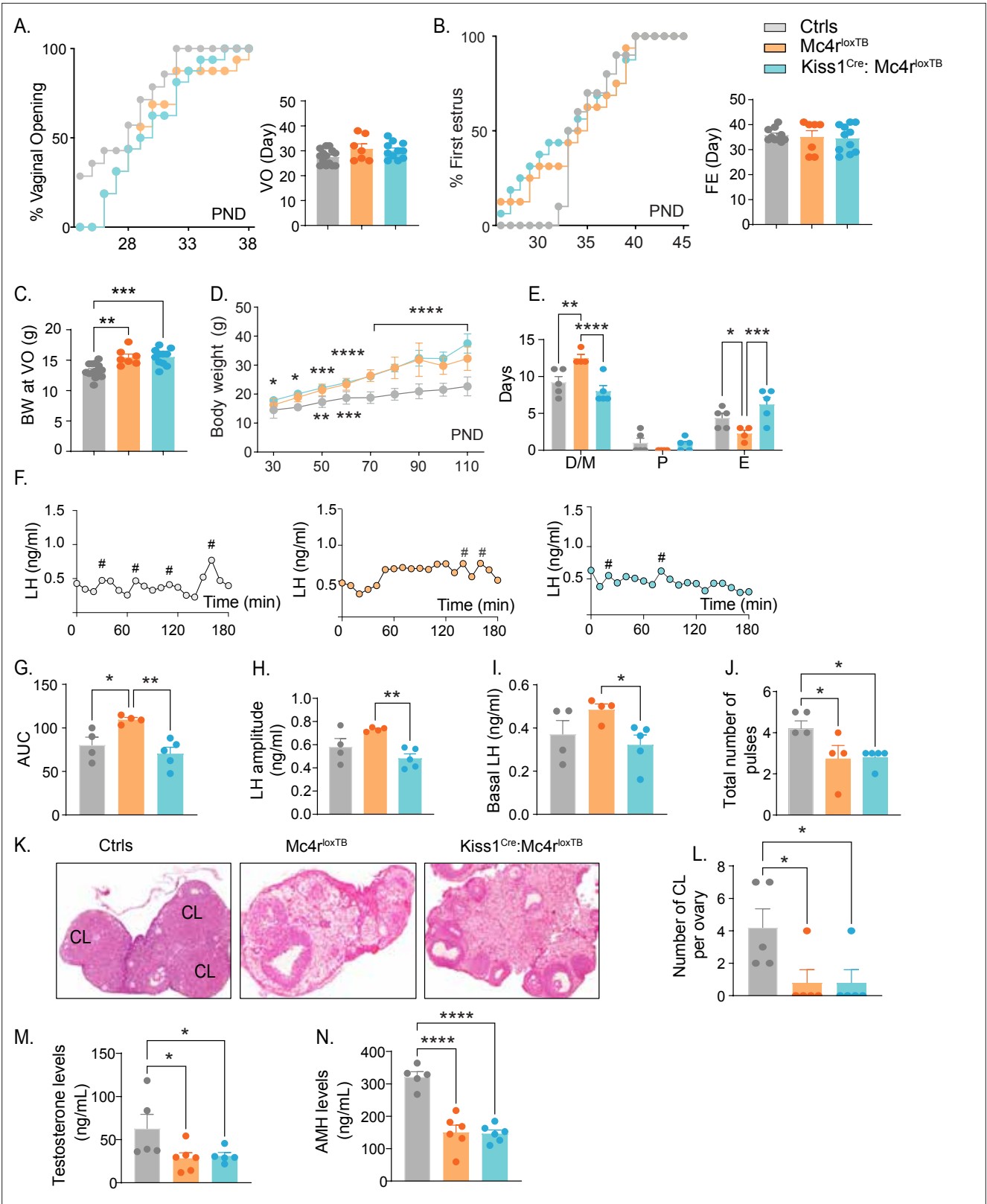

**Figure 3.** Re-insertion of Mc4r in Kiss1 neurons restores estrous cyclicity and luteinizing hormone (LH) pulsatility in *Kiss1^Cre^: Mc4r^loxTB^* females. (**A, B**) *Kiss1^Cre^: Mc4r^loxTB^* (n = 11) and *Mc4r^loxTB^* (Mc4r KO) (n = 7) displayed normal puberty onset as compared to their control littermates (n = 14), as documented by cumulative percent and mean age of animals at vaginal opening and first estrus. (**C**) *Kiss1^Cre^: Mc4r^loxTB^* and *Mc4r^loxTB^* both had significantly higher body weights at the time of puberty onset compared to their controls. **p < 0.01, ***p < 0.001 by one way ANOVA. (**D**) *Mc4r^loxTB^*(n

*Figure 3 continued on next page*

*Figure 3 continued*

= 6) and *Kiss1^Cre: Mc4r^loxTB* (*n* = 11) females displayed significantly higher body weight than their littermates (*n* = 7) from post-natal day 30 onwards. *p < 0.05, **p < 0.01, ***p < 0.001, and ****p < 0.0001. Upper significance (*) represents Ctrls vs. *Kiss1^Cre: Mc4r^loxTB*, and lower significance (*) represents Ctrls versus *Mc4r^loxTB*. Significance was similar between post-natal day (PND) 70 and 110. Data presented as the mean ± SEM. (**E**) *Mc4r^loxTB* females displayed irregular estrous cycles, presenting longer time in diestrus and shorter time in estrus compared to control females, while *Kiss1^Cre: Mc4r^loxTB* females displayed regular estrous cyclicity, similar to controls. *p < 0.05, **p < 0.01, ***p < 0.001, ****p < 0.0001, two-way ANOVA. Data presented as the mean ± SEM. (**F**) Pattern of LH pulsatility analyzed in gonad intact *Mc4r^loxTB* (*n* = 4), *Kiss1^Cre: Mc4r^loxTB* (*n* = 5), and control littermates females (*n* = 5). LH samples were collected every 10 min for 180 min; # represents LH pulses. (**G**) LH total secretory mass, (**H**) LH pulse amplitude, (**I**) basal LH, and (**J**) total number of pulses/180 min were assessed. *p < 0.05, **p < 0.01 by one-way ANOVA. (**K**) Representative samples of ovarian histology from *Mc4r^loxTB*, *Kiss1^Cre: Mc4r^loxTB*, and control females (*n* = 5/group); CL: corpora lutea. Data are presented as the mean ± SEM. (**L**) Ovarian histology showed a significant decrease in the number of corpora lutea in the Mc4r KO and *Kiss1^Cre: Mc4r^loxTB* compared to controls. Groups with different letters are significantly different. Serum levels of (**M**) testosterone and (**N**) anti-Müllerian hormone (AMH) in adult gonad intact females *Mc4r^loxTB*, *Kiss1^Cre: Mc4r^loxTB*, and their control littermates. *p < 0.05, ****p < 0.0001, one-way ANOVA. Data are presented as the mean ± SEM.

The online version of this article includes the following figure supplement(s) for figure 3:

**Figure supplement 1.** Validation of the *Kiss1^Cre: Mc4r^loxTB* mouse model.

so we followed up using a perfusion of the Mc4r-selective agonist THIQ (100 nM) to determine if activation of MC4Rs alone was sufficient to invoke a postsynaptic response in Kiss1^ARH neurons from OVX + E2 females. Similar to MTII, THIQ was able to elicit an excitatory inward current in Kiss1^ARH neurons (4/5 cells, –7.4 ± 0.5 pA), indicating MC4R activation is sufficient (*Figure 4D*).

## POMC neurons synapse directly with Kiss1^ARH neurons

Based on the pharmacological activation of MCRs, we wanted to address whether POMC^ARH neurons are the source of melanocortins to excite Kiss1^ARH neurons directly. To answer this question, we injected an AAV-EF1α-DIO-ChR2:mCherry vector into the ARH of adult *Pomc-Cre* female mice (*Figure 4E*; *Dewailly et al., 2020*). After 2–4 weeks we did whole-cell recordings from putative Kiss1^ARH neurons and looked for a response to high-frequency optogenetic stimulation of POMC fibers in OVX + E2-treated females (*Figure 4F*). We were able to confirm that these were Kiss1 neurons based on the presence of a persistent sodium current and/or *post hoc* identification by scRT-PCR (40/77) cells, (*Figure 4G*). While this current is more prevalent in the AVPV Kiss1 population, Kiss1 neurons are the only ARH neurons to display this electrophysiological 'fingerprint' of a pronounced I_NaP paired with a high capacitance and a low input resistance (*Zhang et al., 2015*). Additionally, while recording from putative Kiss1^ARH neurons, we optogenetically stimulated POMC fibers at low frequency (1 Hz, 5 ms, 470 nm light) and recorded fast postsynaptic inward currents. The fast kinetics of the EPSC are also a tell-tale sign of an ionotropic glutamatergic response (*Clements and Westbrook, 1991*). Previously, CNQX was sufficient to block similar excitatory ESPCs in other postsynaptic targets of POMC neurons (*Stincic et al., 2018*). While the consistent latency from optogenetic stimulus and current response was indicative of a single synaptic delay, we wanted to establish that it was a direct synaptic connection, as we have previously shown for output of POMC neurons (*Stincic et al., 2018*) and Kiss1 neurons (*Qiu et al., 2016*; *Qiu et al., 2018*; *Stincic et al., 2021*). First, we abrogated the optogenetic response with the addition of TTX (1 μM) to the bath, and then we were able to rescue the postsynaptic glutamate response with the addition of the K^+ channel blockers 4-aminopyridine (4-AP; 0.5 mM) and tetraethyl ammonium (TEA; 7.5 mM) (*Figure 4H*). K^+ channel block enables the calcium influx from ChR2 activation in the presynaptic terminals to be sufficient to restore vesicle fusion. Therefore, there appears to be no intervening synapses between POMC and the downstream Kiss1^ARH neurons.

Previously, we found that E2 increases glutamate release from POMC neurons by increasing *Slc17a6*, which encodes Vglut2 (*Stincic et al., 2018*). Therefore, we compared the postsynaptic glutamatergic responses following two optogenetic stimuli (50-ms interval between light flashes) in Kiss1^ARH neurons from OVX + vehicle versus OVX + E2-treated female mice (*Figure 4I*). As anticipated, we found that treatment with E2 increased the probability of glutamate release from POMC neurons onto Kiss1^ARH neurons based on the significant decrease in the paired pulse ratio of the two stimuli (*Herman et al., 2014*; *Figure 4J*). Although in the present study we used an in vivo treatment paradigm, we know from previous studies that this augmentation of glutamate release can happen quite rapidly after a brief exposure to E2 in vitro (within 15 min; *Stincic et al., 2018*). Therefore, Kiss1^ARH neurons are excited by the glutamatergic input from POMC neurons in an E2-dependent manner.

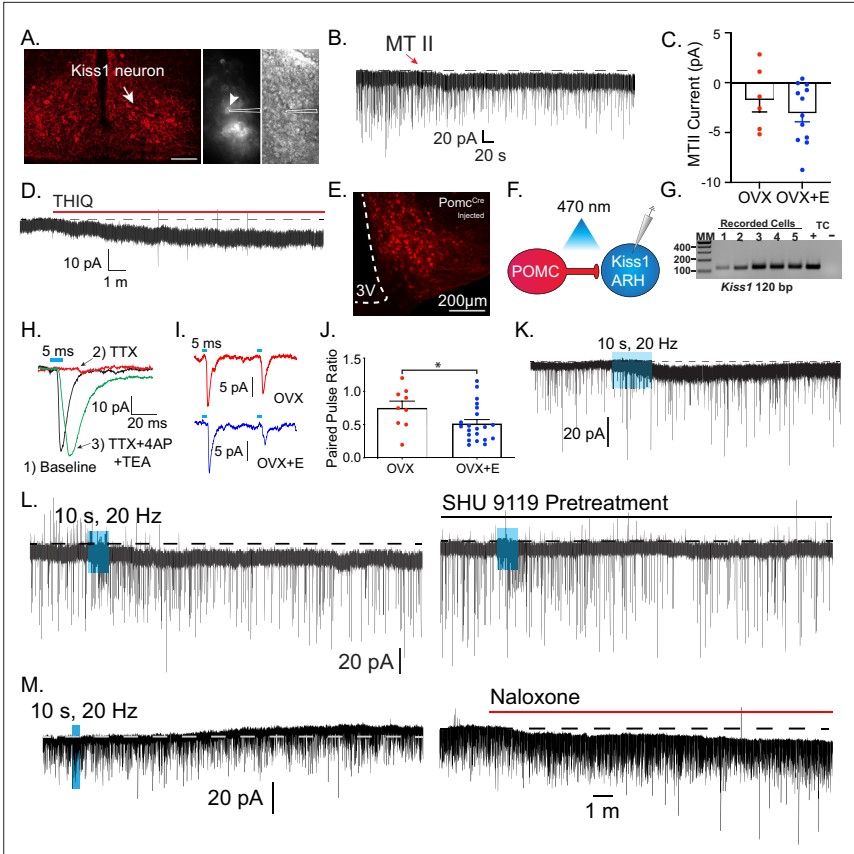

**Figure 4.** Kiss1[ARH] neurons are activated by melanocortin agonists and respond to optogenetic stimulation of POMC neurons. (**A**) Slices were taken from *Kiss1[Cre]*-injected (confocal image) brains to target fluorescent cells (arrowhead) for recording (white outline over electrode). (**B**) Whole-cell voltage-clamp recording of Kiss1[ARH] neurons following the direct application of the high-affinity melanocortin receptor agonist melanotan II (MTII, 250 nM) was added directly to the bath, and the response was compared between slices from ovariectomy (OVX) and OVX + E females. (**C**) While the average inward current was slightly higher in the OVX + E state, there was not a significant difference (Student's *t*-test for unpaired samples, p > 0.05). (**D**) The Mc4r-selective agonist THIQ (100 nM) was perfused, and excitatory inward currents were generated in Kiss1[ARH] neurons. (**E**) Images are of AAV-driven labeling of POMC cells as seen through the confocal. (**F**) Using the AAV-driven expression of channelrhodopsin in adult *Pomc[Cre]* mice, high-frequency stimulation elicited a slow inward current in Kiss1 neurons. (**G**) The identity of cells was confirmed through the presence of a persistent sodium current (see *Zhang et al., 2015*) and/or with RT-PCR of harvested cytoplasm showing *Kiss1* expression (Gel: MM = molecular marker, TC = tissue controls). (**H**) A direct synaptic projection from POMC to Kiss1[ARH] neuron (post hoc identified) was confirmed using the 'rescue' protocol: 1, baseline glutamatergic responses were initially generated (black trace); 2, then action potentials were eliminated by blocking voltage-gated sodium channels with tetrodotoxin (TTX) and the postsynaptic response (red trace); 3, blockade of potassium channels facilitated calcium entry into the terminal through ChR2 to release synaptic vesicles, 'rescuing' the postsynaptic glutamate response (green trace). (**I**) Glutamatergic responses were also often observed, particularly in the ventral ARH. As seen when targeting low input resistance neurons (i.e., Kiss1[ARH]) with low-frequency optogenetic stimulation (5 ms pulse, 50 ms inter-spike interval), the first response in OVX + vehicle females was larger relative to the second response in OVX + E2-treated female mice. Representative traces are the average of 30 sweeps. (**J**) The averaged paired-pulse ratio was lower in recordings from estradiol-treated female mice, indicating an increased release probability (Student's *t*-test *P*<0.05). (**K**) High-frequency optogenetic stimulation elicited a small inward current. (**L**) In a different cell, a high frequency inward current was noted before constant perfusion of SHU9119 for 15 min (break between traces). When the stimulation protocol was repeated, no inward current was elicited. (**M**) The majority of high-frequency responses were inward, and only twice was an inhibitory outward current recorded in identified Kiss1[ARH] neurons. Perfusion of the non-selective opioid receptor antagonist naloxone reversed the current, eliminating MCR activation as the mechanism.

The online version of this article includes the following source data for figure 4:

*Figure 4 continued on next page*

*Figure 4 continued*

**Source data 1.** Tif images (uncropped labeled and unlabeled), showing the original gel picture for *Figure 4G*, indicating the relevant bands.

**Source data 2.** Tif images (uncropped labeled and unlabeled), showing the original gel picture for *Figure 4G*, indicating the relevant bands.

Next, we tested for a melanocortin response using a high-frequency stimulation protocol known to elicit peptide release (*Qiu et al., 2018*). Only cells displaying a glutamatergic response were tested. Indeed, high-frequency stimulation evoked an excitatory inward current (*Figure 4K*) that was blocked by the selective MC3/4R antagonist SHU 9119 (*Figure 4L*). However, the number of high-frequency responses seen in Kiss1 neurons was unexpectedly infrequent and relatively small (4/16, 25%, mean inward current: –1.4 ± 3.1 pA) compared to the efficacy of pharmacological MCR activation. E2 treatment enhances *Pomc* expression and β-endorphin labeling (*Thornton et al., 1994*; *Petersen et al., 1993*), but the effect, if any, on αMSH release or the MCRs in Kiss1 neurons is unknown. We suspected that the high-frequency stimulation was eliciting co-release of αMSH and β-endorphin, which would exert opposite effects on a postsynaptic cell. In order to isolate melanocortin signaling, we pretreated slices with the non-selective opioid antagonist naloxone (1 µM) and found an increase in the likelihood and magnitude of a response to high-frequency stimulation (9/23, 40%, mean inward current: –7.6 ± 2.9 pA) (*Figure 4M*).

Together, these optogenetic findings reinforced our pharmacological results showing that Kiss1$^{ARH}$ neurons are excited by MCR agonists. Additional studies would need to be done to identify the cation conductance that is affected by the MC4R signaling cascade in Kiss1$^{ARH}$ neurons, but clearly, we have established that there are excitatory glutamatergic and peptidergic inputs from POMC to Kiss1$^{ARH}$ neurons.

## Kiss1$^{AVPV/PeN}$ neurons are inhibited by Mc4r agonists

Having established a direct, excitatory projection from POMC to Kiss1$^{ARH}$ neurons, we next examined POMC inputs to Kiss1$^{AVPVPeN}$ neurons. First, we used immunocytochemistry to label fibers expressing αMSH to demonstrate that POMC fibers densely innervated the AVPV/PeN area (*Figure 5A*). Next, we recorded from AVPV neurons in slices taken from OVX + E2, POMC-Cre mice injected with AAV1-DIO-YFP:ChR2 into the ARH (*Figure 5B*). We targeted cells along the ventricle in the AVPV and PeN, surrounded by YFP terminals, and Kiss1$^{AVPV/PeN}$ neurons were identified based on the expression of several endogenous conductances (I$_{NaP}$, T-type calcium current, h-current) that are unique to these neurons (*Zhang et al., 2015*; *Zhang et al., 2013*). We recorded a direct glutamatergic synaptic response following optogenetic stimulation in six neurons, which was further verified through pharmacological 'rescue' of the synaptic response in the majority (5/6) neurons tested (*Figure 5C*), indicating that POMC neurons make direct monosynaptic connections with Kiss1$^{AVPV/PeN}$ neurons.

We next hypothesized that Kiss1$^{AVPV/PeN}$ neurons would be excited by exogenous application of melanocortin agonists because the receptor is typically Gs-coupled (*Mountjoy et al., 1994*; *Yu et al., 2020*). We did whole-cell, voltage-clamp recordings from Kiss1$^{AVPV/PeN}$-Cre::Ai32 neurons, which were isolated synaptically by bathing the neurons in TTX (1 µM), from OVX females. For these experiments, we tested the response to the high-affinity melanocortin receptor agonist MTII (500 nM). Surprisingly, we consistently measured an outward (inhibitory) current that could be reversed on washout of MTII (*Figure 5D*). This outward current was associated with an increase in K$^+$ conductance based on the current-voltage plot (i.e., the outward current reversed at ~E$_{K+}$, *Figure 5E*). Also, the inwardly rectifying I/V could indicate that MC4R is coupled to activation of inwardly rectifying K$^+$ channels, as has been previously reported for MC4R signaling in pre-autonomic parasympathetic neurons in the brainstem (*Sohn et al., 2013*). We hypothesized that the coupling could be modulated by E2 via a Gα$_q$-coupled membrane estrogen receptor (Gq-mER) as we have previously demonstrated in POMC neurons (*Kelly et al., 1992*; *Lagrange et al., 1994*; *Qiu et al., 2003*). For these experiments, we again utilized OVX females and targeted Kiss1$^{AVPV/PeN}$-Cre::Ai32 neurons. Once in the whole-cell voltage-clamp configuration, we perfused the slices with STX (10 nM), a selective ligand for the putative Gq-mER (*Qiu et al., 2003*), and tested the response to the MTII. Indeed, the outward current (inhibitory) response to MTII was completely abrogated and even reversed by STX (*Figure 5F, G*). The short-term (bath) treatment with STX ensured that there was no desensitization of the Gq-mER with longer-term (in vivo)

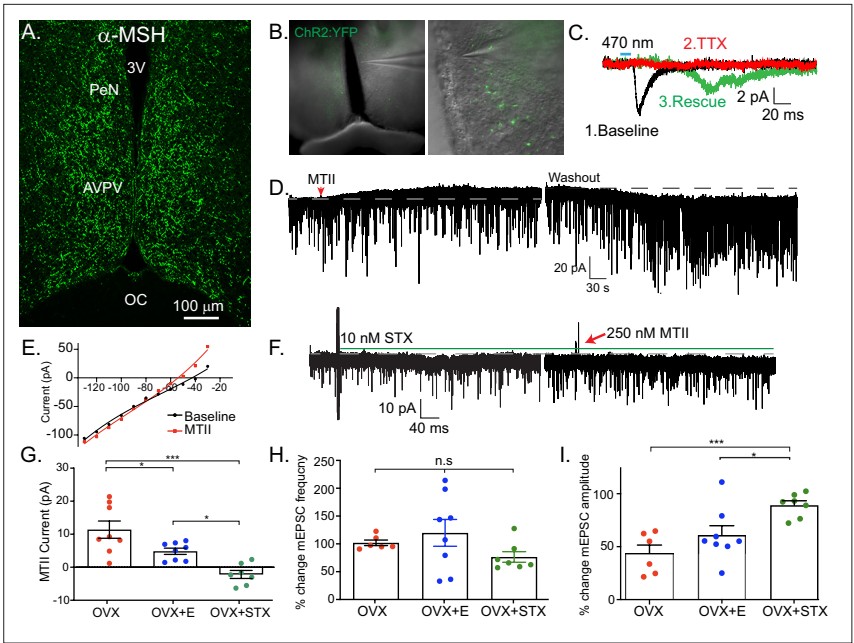

**Figure 5.** Kiss1[AVPV/PeN] neurons are inhibited by Mc4r agonists in an E2-dependent manner. (**A**) Immunohistochemistry showing robust labeling of α-melanocyte stimulating hormone (αMSH) fibers in the AVPV/PeN region in ovariectomy (OVX) + E2 WT female mice. (**B**) Low and high power bright-field images taken during electrophysiology recording in OVX + E2 POMC-Cre mice expressing YFP:ChR2 in the ARH following AAV injection. (Left) Low power image shows the location of recorded cells. (Right) Higher power image of area from white box inset displaying POMC fibers innervating the area. (**C**) Fibers surrounded putative Kiss1[AVPV/PeN] neurons (expressing $I_{NaP}$, $I_T$, and $I_h$), and optogenetic stimuli were able to elicit postsynaptic currents that were eliminated with tetrodotoxin (TTX, 1 µM), but 'rescued' with the addition of K+ channel blockers (4-aminopyridine [4-AP] and tetraethyl ammonium [TEA]), which indicates a monosynaptic connection between POMC[ARH] and Kiss1[AVPV] neurons. (**D**) Whole-cell voltage-clamp recordings were made in ChR2-YFP positive cells in brain slices taken from *Kiss1[Cre]*:Ai32 female mice. Bath application of melanotan II (MTII, 500 nM) generated an inhibitory outward current in a Kiss1[AVPV/PeN] neuron from OVX female. As the cell was synaptically isolated using bath-applied TTX (1 µM), this represents a direct effect. Washout of MTII while still in TTX quickly led to a return to baseline RMP. (**E**) An IV relationship was plotted using voltage steps before and after MTII administration. The crossing at –80 mV (~EK+) indicates that the opening of K+ channels underlies the MC4R inhibition of Kiss1[AVPV/PeN] neurons. (**F**) In a subset of recordings from OVX brain slices, the selective membrane estrogen receptor (Gq-mER) agonist STX (10 nM) was added to the bath for ~10 min prior to the addition of MTII. STX pretreatment resulted in either a greatly attenuated outward current or even an inward current, as shown in this example. (**G**) The mean outward current was significantly higher when recording in Kiss1[AVPV/PeN] neurons from brain slices from vehicle-treated, OVX females compared to E2-treated, OVX females or acute STX-treated brain slices from OVX females (one-way ANOVA $F_{(2,19)}$ = 12.32, p < 0.001; Holm–Sidak post hoc comparisons found significant differences between all groups; *p < 0.05, ***p < 0.001). (**H**) There was no difference in the frequency of mEPSCs (miniature excitatory postsynaptic current) after MTII, calculated as a percent of baseline between groups. (**I**) However, there was a significant difference in the mEPSC amplitude (calculated as a percent of baseline) between all groups: main effect $F_{(2,19)}$ = 10.58, p < 0.001. Post hoc comparisons using Holm–Sidak found OVX + STX to be different from both OVX (p < 0.001) and OVX + E (p < 0.05).

treatment with E2. Therefore, estrogen receptor activation can rapidly uncouple (i.e., desensitize) the MC4R inhibitory response in Kiss1[AVPV/PeN] neurons. We would predict that the intracellular signaling cascade for the heterologous desensitization is similar to what we have elucidated in POMC neurons (*Qiu et al., 2003*), but this will need to be determined in future experiments. We next investigated the response in vehicle-treated, OVX females. Indeed, Kiss1[AVPV/PeN] neurons were even more inhibited by the same MTII exposure (*Figure 5G*). Therefore, in contrast to Kiss1[ARH] neurons, the MC4R appears to be coupled to the opening of K+ channels in Kiss1[AVPV/PeN] neurons. To further establish whether the MTII was having pre- or postsynaptic effects, we measured the glutamatergic mEPSCs before and after melanocortin receptor activation. We found there was no significant effect on the frequency

(*Figure 5H*), but there was an effect on the amplitude (*Figure 5I*). This further supports a postsynaptic locus of STX's effects. However, post hoc comparisons found STX increased amplitude compared to both OVX and OVX + E. This result hints at a membrane-delimited effect as STX is a much more potent agonist for Gq-mER than E2 (*Qiu et al., 2006*).

## Discussion

Our findings show that melanocortins act directly on Kiss1 neurons to contribute to the metabolic regulation of fertility, in line with previous reports showing that αMSH stimulates LH release in a kisspeptin-dependent manner (*Manfredi-Lozano et al., 2016*) and that Kiss1 neurons express the melanocortin receptor MC4R (*Cravo et al., 2011*; *Lam et al., 2021*; *Villa et al., 2024*; *Stephens and Kauffman, 2021*). Several studies in humans have linked *MC4R* mutations to reproductive abnormalities, including precocious puberty (*Doulla et al., 2014*), PCOS (*Batarfi et al., 2019*), and hypogonadism (*Hainerová et al., 2011*). However, a direct association between *MC4R* mutations and reproductive function, independent from the obese condition of these patients, has not been identified (*Farooqi et al., 2003*). In the current study, we show that the reproductive impairments observed in Mc4r-deficient mice, which replicate many of the conditions described in humans, are largely mediated by the direct action of melanocortins via Mc4r on Kiss1 neurons and not to their obese phenotype. This is because the ablation of Mc4r from Kiss1 neurons largely replicated the reproductive impairments observed in Mc4r KO female mice without inducing obesity, and the selective re-insertion of Mc4r into Kiss1 neurons of Mc4r KO mice significantly improved their reproductive function without changing their obese phenotype. Strikingly, puberty onset was advanced in Kiss1- Mc4r KO females. Our findings revealed low melanocortin expression in the hypothalamus of WT females during pubertal development, characterized by lower levels of *Agrp*, *Mc4r*, and *Mc3r* expression at the time of puberty onset (~PND30). This developmental decline in the expression of melanocortin receptors is in line with the advanced timing of puberty onset in females when Mc4r is congenitally ablated from Kiss1 neurons. A recent study using whole body MC3RKO mice showed a trend to delayed puberty onset (*Lam et al., 2021*), suggesting a larger role of melanocortins in the timing of puberty onset. Here, we show that in females, the combination of MC3R and MC4R action may be necessary for pubertal development. Interestingly, the *Mc4r*[loxTB] and *Kiss1*[Cre]: *Mc4r*[loxTB] mice do not show an advancement in puberty onset as we observed in Kiss1- Mc4r KO mice; however, these mice are obese at the time of puberty onset already, which may affect the timing of sexual maturation independently of MC4R action in Kiss1 neurons. Of note, the findings presented in this study do not rule out the existence of additional sites of action of MC4R in other neuronal populations to regulate reproduction, for example, GnRH neurons, which also express Mc4r (*Israel et al., 2012*; *Roa and Herbison, 2012*). Indeed, the lack of full recovery of the reproductive function in *Kiss1*[Cre]: *Mc4r*[loxTB] mice suggests that Mc4r in Kiss1 neurons is necessary but not sufficient to achieve full reproductive capabilities.

In addition to early puberty onset, Kiss1- Mc4r KO females displayed an impaired preovulatory LH surge driving ovulation and had fewer corpora lutea in their ovaries compared to control littermates, further supporting a role of MC4R in regulating ovulation. While Mc4r has been documented in the ovary (*Chen et al., 2017*), our data of (1) impaired LH surge observed in the Kiss1- Mc4r KO females (*Figure 2M*), and (2) direct action of αMSH on Kiss1 ARH and AVPV/PeN neuronal populations through Mc4r (*Figures 4 and 5*) support a CNS role for MC4R in the regulation of fertility. However, an additional effect of MC4R at the level of the ovary cannot be ruled out. One possibility is that the genetic deletion of *Mc4r* from the *Kiss1* gene may have affected the ovary. However, to test this hypothesis, the co-expression of *Mc4r* and *Kiss1* in the ovary needs to be investigated. Another possibility is that, given the partial fertility of these mice, follicles may have developed at different rates, potentially due to irregular secretion of gonadotropins. As a result, fewer follicles would develop and ovulate. In adulthood, the reproductive phenotype observed in Kiss1- Mc4r KO and *Mc4r*[loxTB] female mice (increased LH, irregular estrous cycles, oligo-ovulation, increased cystic follicles) correlates with the phenotype observed in PCOS mouse models (*Moore et al., 2013*; *McCarthy et al., 2022*). In fact, an association between *MC4R* mutations and PCOS has been reported (*Batarfi et al., 2019*). However, our mouse models failed to display higher levels of circulating androgens or AMH, two of the hallmarks of PCOS (*Dewailly et al., 2020*), suggesting that melanocortin signaling is unlikely to be a main contributing factor in the development of this syndrome.

Our data revealing early puberty onset, augmented LH pulse, and impaired LH surge driving ovulation support a direct role of MC4R on the two primary Kiss1 populations regulating these functions (ARH and AVPV/PeN). We further investigated this and found that Mc4r signaling excites Kiss1[ARH] neurons and inhibits Kiss1[AVPV/PeN] neurons in an estradiol-dependent manner. In Kiss1[ARH] neurons, bath-applied or optogenetically evoked release of melanocortin agonists induces a direct excitatory inward current. Our studies using whole-cell voltage clamp in female mice were able to detect a melanocortin-mediated inward current in Kiss1[ARH] neurons that would increase excitability. Previously, an MCR-activation effect was not detected in cell-attached recordings of diestrus females (*Manfredi-Lozano et al., 2016*; *Sabine Hessler and Herbison, 2020*). However, cell-attached recordings do not measure a direct response in isolated Kiss1[ARH] neurons but rather the summation of multiple synaptic inputs to Kiss1[ARH] neurons. Most importantly, we did not see a difference in the excitatory response in synaptically isolated Kiss1[ARH] neurons from E2-treated versus vehicle-treated, ovariectomized females. This emphasizes the importance of doing whole-cell recording from isolated Kiss1 neurons. The modulatory actions of POMC[ARH] neuronal projections to Kiss1[ARH] neurons can also occur through (1) the excitatory action of glutamate in a process that is facilitated in the presence of estradiol, in line with previous publications (*Qiu et al., 2018*; *Stincic et al., 2018*), and (2) the inhibitory action of β-endorphin, suggesting that POMC[ARH] neurons are able to exert a precise regulatory role of the GnRH pulse generator, that is Kiss1[ARH] neurons, in response to metabolic challenges. In the AVPV/PeN, Mc4r agonists inhibited Kiss1[AVPV/PeN] neurons in an estradiol-dependent manner, which utilized a $G\alpha_q$-coupled membrane estrogen receptor (STX receptor) to attenuate the inhibitory tone of Mc4r in these neurons. Rapid estrogen signaling may act to ease transitions between states. Membrane-delimited E2 actions can quickly attenuate or enhance coupling between receptors and signaling cascades. These effects will precede E2-driven changes in gene expression that produce more stable alterations in signaling. This combination of mechanisms will reduce any lag between rises in serum E2 and physiological effects. Considering the abbreviated mouse reproductive cycle, parallel mechanisms acting on different timescales are particularly important.

Our pharmacology data clearly demonstrate the estradiol-dependent action of Mc4r in Kiss1[AVPV/PeN] neurons, suggesting that similar to Kiss1[ARH] neurons, this population is regulated by melanocortins. The effect of Mc4r agonists on Kiss1 neurons in the presence of TTX suggests a direct synaptic effect without the need for channelrhodopsin-2 (ChR2)-assisted circuit mapping. Nonetheless, to further demonstrate this interaction, we provide evidence of profound innervation of αMSH fibers and the presence of synaptic contact between POMC and Kiss1[AVPV/PeN] neurons through the optogenetic stimulation of a glutamatergic response in Kiss1[AVPV/PeN] neurons after low-frequency stimulation of POMC terminals. We intentionally avoided high-frequency stimulation of POMC terminals to prevent the co-release of β-endorphins (e.g., *Figure 4M*) and their possible additive effect on the inhibitory action of MC4R. Future studies will be aimed at the characterization of the endogenous opioid pathways in the induction of the LH surge.

To date, two instances of MC4R-mediated inhibition have been described in the central nervous system: MC4R can Gi,o couple to open K+ (Kir6.2) channels in parasympathetic preganglionic neurons in the brainstem (*Sohn et al., 2013*), and MC4R can directly couple to K+ (Kv7.1) channels in the hypothalamic paraventricular nucleus neurons (*Ghamari-Langroudi et al., 2015*). Interestingly, our findings describe an inhibitory role of MC4R in the reproductive neuroendocrine axis. This steroid state-dependent effect is particularly relevant because, in the context of reproduction, the two populations of Kiss1 neurons are strikingly different in their response to estradiol, where peptide expression in Kiss1[ARH] and Kiss1[AVPV/PeN] neurons is inhibited or stimulated, respectively, in order to mount the negative versus positive feedback of sex steroids (*Goodman et al., 2022*). This differential regulation allows for the episodic versus surge release of GnRH; however, the cellular mechanisms underlying these opposing roles of Kiss1 neurons to the same stimulus, that is circulating E2 levels, remain unknown. Although there were no differences in the magnitude of the electrophysiological responses to MTII in Kiss1[ARH] neurons in vehicle- versus E2-treated, ovariectomized females, we cannot rule out that E2 affects other systems activated by the MC4R signaling cascade as recently reported in MC4R-expressing neurons of the ventromedial nucleus of the hypothalamus (*Krause et al., 2021*). Interestingly, puberty onset was advanced in Kiss1- Mc4r KO females. This data suggests that the inhibitory tone of MC4R signaling on Kiss1[AVPV/PeN] neurons might also be involved in the timing of puberty onset, which in turn suggests a role for this female-specific population of Kiss1 neurons in

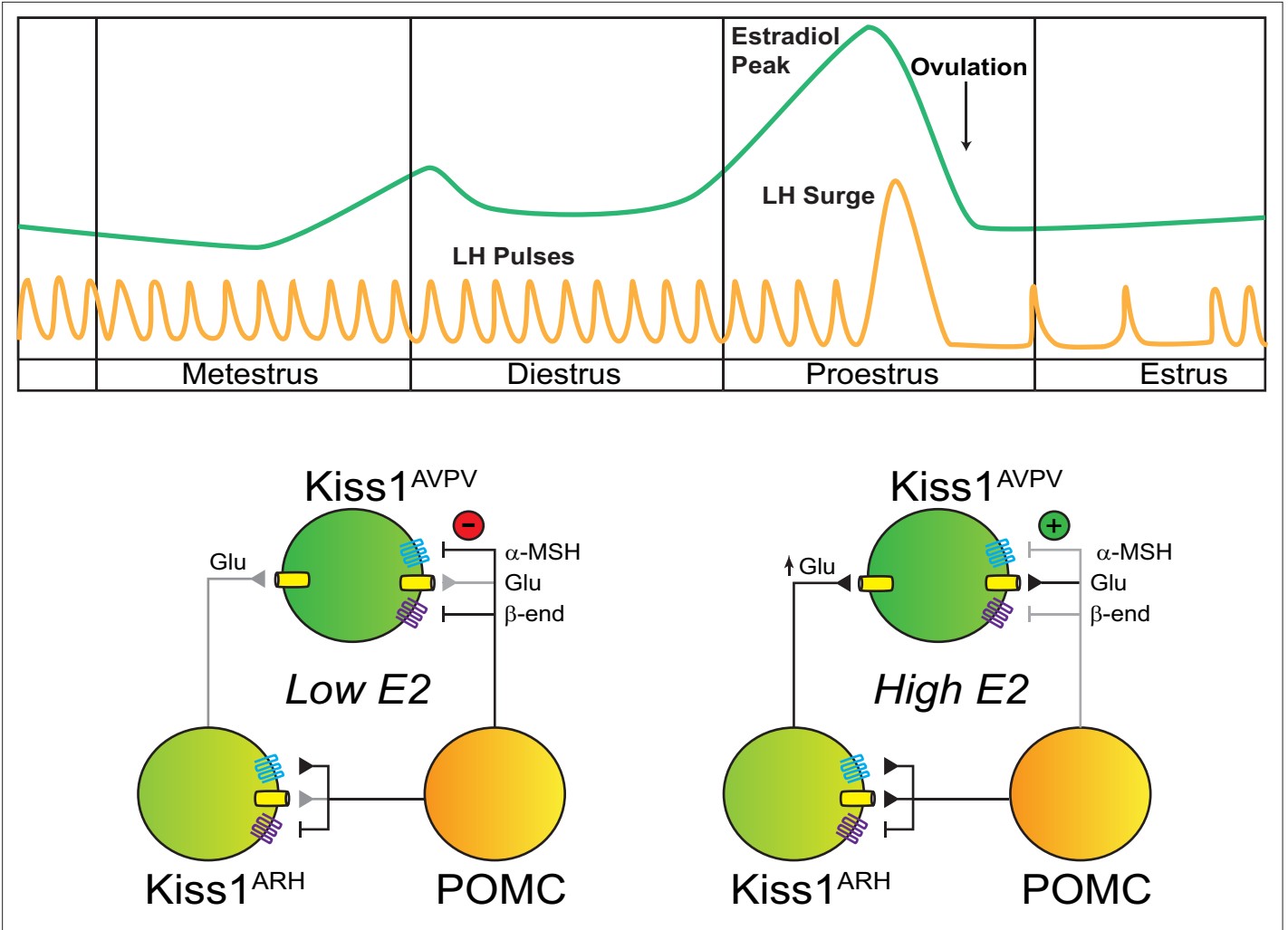

**Figure 6.** Working model of the neuronal input from POMC ARH to Kiss1 ARH and AVPV/PeN populations. POMC[ARH] neurons excite Kiss1[ARH] neurons via glutamate and α-melanocyte stimulating hormone (αMSH) release, which activate glutamatergic ionotropic receptors and Mc4r, respectively. In the low E2 state, αMSH fibers project to the AVPV/PeN, and Mc4r inhibit Kiss1[AVPV/PeN] neurons via opening a K$^+$ channel. However, in the high E2 state, E2 enhances the glutamatergic excitation of both POMC and Kiss1 ARH neurons. E2 treatment rapidly abrogates (uncouples) the inhibitory MC4R signaling in Kiss1[AVPV/PeN] neurons, allowing greater excitatory actions of Kiss1[ARH] neurons via enhanced glutamate release (*Zhang et al., 2015*).

sexual maturation. Furthermore, our findings suggest that metabolic cues, through the regulation of the melanocortin output onto Kiss1[AVPV/PeN] neurons, are essential for the timing and magnitude of the GnRH/LH surge. This study suggests that the melanocortin–kisspeptin pathway is one of the multiple pathways essential for the metabolic regulation of the HPG axis. Future studies are warranted to fully elucidate synaptic input and postsynaptic cascades in the POMC-Kiss1[AVPV/PeN] circuit.

Altogether, our data reveal a differential regulatory action of MC4R in the neural control of GnRH/LH release, participating in both the surge (E2-treated, ovariectomized female) and the pulse-like (ovariectomized female) modes of LH release through the cellular regulation of Kiss1 neurons, which translate into our findings on the conditional knockout of *Mc4r* from Kiss1 neurons (*Figure 6*). These findings are important because the reproductive abnormalities often attributed to obesity in MC4R-deficient patients may not be caused by the excess in body weight but, at least in part, by a deficiency in MC4R signaling directly at the level of Kiss1 neurons that affects predominantly the ability to mount a preovulatory LH surge.

## Materials and methods

### In vivo experimental procedures

#### Generation of Kiss1- Mc4r KO, *Kiss1*^Cre: *Mc4r*^loxTB, and *Mc4r*^loxTB transgenic mice

Kiss1- Mc4r KO mice were generated by crossing *Kiss1*^Cre knock-in mice and *Mc4r*^lox/lox mice. *Kiss1*^Cre mice (RRID:MGI:6278139) were obtained from Dr. Richard Palmiter (University of Washington, Seattle, WA) (*Padilla et al., 2018*) and *Mc4r*^lox/lox (RRID:IMSR_JAX:023720) were a gift from Dr. Brad Lowell (Beth Israel Deaconess Hospital, Boston, MA; *Shah et al., 2014*). These mice were crossed to generate Kiss1- Mc4r KO mice lacking *Mc4r* expression selectively from Kiss1 neurons (*Kiss1*^Cre +/; *Mc4r*^lox/lox) and their control littermates (Mc4r^lox/lox). To generate *Kiss1*^Cre: *Mc4r*^loxTB and *Mc4r*^loxTB mice, *Mc4r*^loxTB mice were purchased from The Jackson Laboratory (*Mc4r*^loxTB; catalog no. 006414). These mice present with a loxp-flanked transcriptional blocking (LoxTB) sequence preventing normal endogenous gene transcription and translation from the endogenous locus. Homozygous *Mc4r*^loxTB mice are devoid of functional Mc4r mRNA (Mc4r KO mice), while the presence of Cre recombinase on the Kiss1 promoter in *Kiss1*^Cre mice will result in the removal of the transcription blocker and subsequent expression of Mc4r in tissue-specific sites (i.e., Kiss1 neurons), therefore resulting in the generation of *Kiss1*^Cre: *Mc4r*^loxTB (with *Mc4r* expression restored in Kiss1 neurons), their obese control littermates *Mc4r*^loxTB mice (Mc4r KO), and their WT controls (*Mc4r*^+/+). To rule out early embryonic recombination of the *Mc4r*^loxTB/loxTB or *Mc4r*^lox/lox alleles, we ran PCR assays on tail DNA designed to detect wild-type allele (*Mc4r*^+/+), undeleted lox (*Mc4r*^lox/lox) or loxTB alleles (*Mc4r*^loxTB/loxTB). Genotyping was confirmed by sending tissue to Transnetyx, Inc, for testing by real-time polymerase chain reaction. Mice were housed in Harvard Medical School Animal Resources facilities where they were fed standard mouse chow (Teklad F6 Rodent Diet 8664) and were given ad libitum access to tap water under constant conditions of temperature (22–24°C) and light (12 hr light [07:00]/dark [19:00] cycle). For each experiment, mice of the same litter and similar age litters were randomly allocated into groups after genotyping. The sample size of mice is specified in each figure legend, according to the experimental approach used.

#### RNAscope in situ hybridization

To validate the Kiss1- Mc4r KO, *Kiss1*^Cre: *Mc4r*^loxTB, and *Mc4r*^loxTB (Mc4r KO) mouse models and investigate the co-expression of *Kiss1* and *Mc4r* mRNA in these mice, dual fluorescence ISH was performed using RNAscope (ACD, Multiplex Fluorescent v.2) according to the manufacturer's protocol using the following probes: *Mc4r* (319181-C2) and *Kiss1* (500141-C1). Brains (n = 4/group) from WT OVX (for expression in the ARH) and OVX + E2 (for expression in the AVPV/PeN) mice and OVX Kiss1- Mc4r KO, *Kiss1*^Cre: *Mc4r*^loxTB and Mc4r KO were removed fresh frozen on dry ice, and then stored at –80°C until sectioned. Five sets of 20 μm sections in the coronal plane were cut on a cryostat, from the diagonal band of Broca to the mammillary bodies, thaw mounted onto SuperFrost Plus slides (VWR Scientific) and stored at –80°C until use. A single set was used for the ISH experiment (adjacent sections 100 mm apart). Images were taken at ×20 magnification of the sections containing AVPV, PeN, and the three rostro-to-caudal levels of the ARH, and Kiss1 neurons expressing (*Kiss1*^Cre: *Mc4r*^loxTB mice) or lacking (Kiss1- Mc4r KO mice) *Mc4r* were identified using ImageJ.

#### Reproductive maturation of Kiss1- Mc4r KO, *Kiss1*^Cre: *Mc4r*^loxTB, and *Mc4r*^loxTB mice

To assess the reproductive phenotype of mice with selective re-insertion of *Mc4r* in Kiss1 neurons (*Kiss1*^Cre: *Mc4r*^loxTB, n = 19), selective deletion of *Mc4r* from Kiss1 neurons (Kiss1- Mc4r KO, n = 24), global deletion of *Mc4r* (*Mc4r*^loxTB, n = 7); and their control Mc4r^lox/lox littermates (n = 14), mice were weaned at PND21 and were monitored daily for puberty onset. Females were monitored daily for VO (indicative of the complete canalization of the vaginal cavity) and for FE (first day with cornified cells determined by daily morning vaginal cytology) after the day of VO. Body weight (BW) was measured on the day of puberty onset to determine if changes in puberty onset could be due to differences in BW. Estrous cyclicity was monitored in females by daily vaginal cytology, for a period of 14–15 days, in 6-month-old mice and their respective control littermates (n = 5/group). Cytology samples were obtained every morning (9 a.m.), placed on a glass slide and stained with hematoxylin and eosin for determination of the estrous cycle stage under the microscope.

## Fecundity test of Kiss1- Mc4r KO females

Adult 6-month-old Kiss1- Mc4r KO and control littermate female mice were placed with adult WT males proven to father litters for 90 days and time to deliver pups and number of pups per litter were monitored.

## Characterization of the estradiol-induced LH surge

Kiss1- Mc4r KO ($n$ = 6) and control $Mc4r^{lox/lox}$ littermate ($n$ = 5) adult female mice were subjected to bilateral OVX via abdominal incision under light isoflurane anesthesia. Immediately after OVX, capsules filled with E2 (1 µg/20 g BW) were implanted subcutaneously via a small mid-scapular incision on the back. Five days later, mice were subcutaneously injected in the morning with estradiol benzoate (1 µg/20 g BW) to produce elevated proestrus-like E2 levels (induced LH surge) on the following day (*Dror et al., 2013*). Blood samples were collected at 8 a.m. and 7 p.m.; LH levels were stored at –80°C until measured via LH ELISA.

## Ovarian histology and hormone measurements

Bilateral OVX of 6-month-old Kiss1- Mc4r KO ($n$ = 4), $Kiss1^{Cre}$: $Mc4r^{loxTB}$, $Mc4r^{loxTB}$ and their control littermates ($n$ = 5/group) was performed under light isoflurane anesthesia. Briefly, the ventral skin was shaved and cleaned, and one small abdominal incision was made. Once the ovaries were identified and excised, the muscle incision was sutured, and the skin was closed with surgical clips. Ovaries were stored in Bouin's fixative, sectioned, and stained with hematoxylin and eosin at the Harvard Histopathology Core. Corporal lutea were counted in the middle section of each ovary. Serum samples were also collected for analysis of testosterone and AMH levels in these mice. These hormone levels were measured at the University of Virginia Ligand Assay core with the Mouse & Rat Testosterone ELISA assay (reportable average range 10–1600 ng/dl; sensitivity of 10 ng/dl); AMH ELISA assay (reportable average range 0.2–15 ng/ml; sensitivity of 0.2 ng/ml).

## LH pulsatile secretion profile in gonad intact Kiss1- Mc4r KO, $Kiss1^{Cre}$: $Mc4r^{loxTB}$, and $Mc4r^{loxTB}$ female mice

To assess the profile of LH pulses secretion, adult 6 months old gonad intact Kiss1- Mc4r KO females ($n$ = 6), their control $Mc4r^{lox/lox}$ littermates ($n$ = 7); and $Kiss1^{Cre}$: $Mc4r^{loxTB}$ ($n$ = 5/group), $Mc4r^{loxTB}$ and their control WT females ($n$ = 4/group) were handled daily for 3 weeks to allow acclimation to sampling conditions prior to the experiment. Pulsatile measurements of LH secretion were assessed in diestrus by repeated blood collection through a single incision at the tip of the tail. The tail was cleaned with saline, and 4 µl of blood was taken at each time point from the cut tail with a pipette. We collected sequential blood samples every 10 min over a 180-min sampling period. Samples were immediately frozen on dry ice and stored at −80°C until analyzed with LH ELISA as previously described (*Steyn et al., 2013*). The functional sensitivity of the ELISA assay was 0.0039 ng/ml with a CV% of 3.3%.

## LH pulses analysis

LH pulses in mice were analyzed using a custom-made MATLAB-based algorithm. The MATLAB code includes a loop that determines LH pulses as any LH peak: (1) whose height is 20% greater than the heights of the two previous values; (2) 10% greater than the height of the following value; and (3) the peak at the second time interval needs to be 20% greater than the single value that comes before it to be considered a pulse, as we previously described (*Talbi et al., 2021*).

LH pulsatility was assessed by measuring: (1) the total secretory mass, assessed by AUC; (2) the LH pulse amplitude, calculated by averaging the four highest LH values in the samples collection period for each animal; (3) the basal LH, calculated by averaging the four lowest LH values in the samples collection period for each animal; and (4) the total number of pulses throughout the 180 min sampling period.

## Immunohistochemistry

### Animals and treatment

Coronal brain blocks (2 mm each) from adult female C57BL/6 mice ($n$ = 4) were fixed by immersion in 4% paraformaldehyde for ~8 hr, cryoprotected in 30% sucrose solution, frozen in isopentane at –55°C,

sectioned coronally on a cryostat at 20 µm, and thaw-mounted onto Superfrost Plus slides (Fisher Scientific, Pittsburgh, PA). The 20 µm sections were stored at –20°C until used for immunocytochemistry.

## Immunocytochemistry

The sections were rinsed in PB (0.1 M phosphate buffer, pH 7.4) for at least 30 min. Next, sections were incubated with normal serum corresponding to the host for the secondary antiserum (5% normal serum with 0.3% Triton X-100 in PBS for 30 min), rinsed in PB, and then incubated for ~45 hr at 4°C with a rabbit polyclonal antiserum against αMSH (1:2500). The specificity of this antiserum has been documented (*Dave et al., 1985*). After rinsing, sections were first incubated for 2–3 hr at room temperature with biotinylated donkey anti-rabbit gamma globulin (IgG; 1:500) and next with streptavidin-Alexa 488 (1:2500). Both the primary and secondary antibodies were diluted in tris-(hydroxymethyl) aminomethane (0.5 %, Jackson ImmunoResearch, Philadelphia, PA) in PB containing 0.7% seaweed gelatin (Jackson ImmunoResearch, Philadelphia, PA) and 0.5% Triton X-100 and 3% bovine serum albumin (Jackson ImmunoResearch, Philadelphia, PA) adjusted to pH 7.6. Following a final rinse overnight, slides were coverslipped with gelvatol containing the anti-fading agent, 1,4-diazabicyclo(2,2)octane (Cold Spring Harbor Protocols, 2006).

## Imaging

Photomicrographs of labeling were initially acquired using a Nikon E800 fluorescent microscope (Eclipse E800; Nikon Instruments, Melville, NY) equipped with a fiber illuminator (Intensilight C-HGFI; Nikon Instruments) and a high-definition digital microscope camera head (DS-Fi1; Nikon Instruments) interfaced with a PC-based camera controller (DS-U3; Nikon Instruments).

## Real-time quantitative PCR

(1) To investigate the changes in the expression of the melanocortin genes *Agrp*, *Pomc*, *Mc3r*, and *Mc4r* in the MBH during development in prepubertal and pubertal WT females at ages P10 ($n = 6$), P15 ($n = 6$), P22 ($n = 6$), and P30 ($n = 5$), and (2) evaluate the gene expression profile of *Pdyn*, *Kiss1*, *Tac2*, *Tacr3*, in the ARH of adult intact (in diestrus) female Kiss1- Mc4r KO ($n = 6$) and their control Mc4r$^{lox/lox}$ littermates ($n = 3$). The brains were removed and rapidly embedded in Tissue-Tek, frozen in −30°C isopentane solution and stored at −80°C until use. Frozen tissue punches were recovered through MBH with a 1 mm diameter canula (*Naulé et al., 2015*). These tissue punches encompassed the whole MBH from the WT females (*Figure 1D*) and the ARH from the Kiss1- Mc4r KO and their control females (*Figure 2I*). The tissues were homogenized, and total RNA was isolated using TRIzol reagent (Invitrogen) followed by chloroform/isopropanol extraction. RNA purity and concentration were measured via an absorbance spectrophotometer (260/280 nm >1.8; NanoDrop 1000, Thermo Fisher Scientific). Total RNA (1 µg) was reverse transcribed to cDNA using random hexamers (High-Capacity cDNA Synthesis Kit, Life Technologies). Quantitative real-time PCR assays were performed using SYBR Green RT-qPCR master mix (Applied Biosystems) and analyzed using ABI Prism 7000 SDS software (Applied Biosystems). The cycling conditions were: 2-min incubation at 95°C (hot start), 45 amplification cycles (95°C for 30 s, 60°C for 30 s, and 45 s at 75°C, with fluorescence detection at the end of each cycle), followed by melting curve of the amplified products obtained by ramped increase of the temperature from 55 to 95°C to confirm the presence of single amplification product per reaction. PCR specificity was verified by melting curve analysis and agarose gel electrophoresis. Each sample was run in duplicate to obtain an average cycle threshold (CT) value, and relative expression of each target gene was determined using the comparative Ct method (*Pfaffl, 2001*). The data were normalized to Hypoxanthine Guanine Phosphoribosyltransferase (*Hprt*) expression levels in each sample. Results were expressed as fold differences in relative gene expression with respect to (1) P10 for melanocortin genes expression analysis during development in WT mice, and (2) controls for the KNDy genes expression in female Kiss1- Mc4r KO mice. The primers used are listed in *Table 1*.

## Data analysis

All behavioral analyses were performed blind to the to the genotype or group. Statistical analyses were performed using GraphPad Prism. Statistical outliers were determined using GraphPad Prism. Statistical data are expressed as means ± SEM, where *n* represents the number of animals in each study group. The significance of differences between groups was evaluated using unpaired Student's

**Table 1.** Primers.

| Gene | Gene length (pb) | Accession # | Primers | Location (nt) | Sequence |
|---|---|---|---|---|---|
| Hprt | 352 | NM_013556.2 | Hprt-F | 704–728 | CCTGCTGGATTAC ATTAAAGCGCTG |
| | | | Hprt-R | 377–401 | GTCAAGGGCATAT CCAACAACAAAC |
| Agrp | 136 | NM_001271806.1 | Agrp-F | 466–488 | GCCTCAAGAAGAC AACTGCAGAC |
| | | | Agrp-R | 580–601 | AAGCAGGACTC GTGCAGCCTTA |
| Pomc | 138 | NM_001278584.1 | Pomc-F | 187–208 | CCATAGATGTGTG GAGCTGGTG |
| | | | Pomc-R | 303–324 | CACCTCCGTTG CCAGGAAACAC |
| Mc4r | 101 | NM_016977.4 | Mc4r-F | 552–570 | CCCGGACGGA GGATGCTAT |
| | | | Mc4r-R | 632–652 | TCGCCACGATC ACTAGAATGT |
| Pdyn | 200 | NM_018863.4 | Pdyn-F | 45–64 | ACAGGGGGAGA CTCTCATCT |
| | | | Pdyn-R | 223–244 | GGGGATGAATG ACCTGCTTACT |
| Kiss1 | 129 | AF472576.1 | Kiss1-F | 147–166 | GCTGCTGCTT CTCCTCTGTG |
| | | | Kiss1-R | 256–275 | TCTGCATACC GCGATTCCTT |
| Tac2 | 234 | NM_001199971.1 | Tac2- F | 238–257 | GCTCCACAG CTTTGTCCTTC |
| | | | Tac2- R | 452–471 | GCTAGCCTTG CTCAGCACTT |
| Tacr3 | 159 | NM_021382.6 | Tacr3-F | 759–779 | GCCATTGC AGTGGACAGGTAT |
| | | | Tacr3-R | 898–917 | ACGGCCTG GCATGACTTTTA |

$t$-test, or a one- or two-way ANOVA test (with *post hoc* comparisons). Significance level was set at p < 0.05. All analyses were performed with GraphPad Prism Software, Inc (San Diego, CA).

## In vitro experimental procedures

### Animals

The animal studies were approved by the Brigham and Women's Hospital Institutional Animal Care and Use Committee (IACUC) in the Center for Comparative Medicine. Adult wild-type (WT) C57/BL6 female mice were group housed under constant conditions of temperature (22–24°C) and light (12:12 hr light:dark cycle), fed with standard mouse chow and ad libitum access to tap water. All animal procedures described in the electrophysiology studies were performed in accordance with institutional guidelines based on National Institutes of Health standards and approved by the Institutional Animal Care and Use Committees at Oregon Health and Science University and Appalachian State University. *Kiss1^Cre* (v2) mice (*Padilla et al., 2018*) were crossed with Ai32 (*Madisen et al., 2012*) or C57B6J mice. *Pomc-Cre* mice (RRID:IMSR_JAX:005965; *Balthasar et al., 2004*) were crossed with wild-type C57B6J (RRID:IMSR_JAX:000664) mice. The Ai32 cross was not used with *Pomc-Cre* mice because the gene is transiently expressed during development in some cells fated to be Kiss1 or AgRP cells (*Padilla et al., 2010*), and early recombinant events lead to persistent expression of the ChR2-mCh fusion protein in non-POMC cells. However, we have previously shown that AAV-driven expression in adult *Pomc-cre* animals is restricted to β-endorphin labeled cells in the ARH (***Stincic***

*et al., 2018*), avoiding this non-specificity issue. All colonies were maintained onsite under controlled temperature (21–23°C) and photoperiod (12:12 hr light–dark cycle 0600–1800) while receiving ad libitum food (5L0D; LabDiet, St. Louis, MO) and water access. Following surgeries, mice received a *s.c.* dose of 4–5 mg/kg carprofen (Rimadyl; Pfizer Animal Health, New York, NY) and given at least 1 week of recovery.

## Ovariectomies

Seven to ten days prior to each experiment, ovaries were removed as described previously while under isoflurane anesthesia (*Stincic et al., 2018*). Two days before experiments, females received either an injection of sesame oil (50 µl, *sc*; Sigma-Aldrich, St. Louis, MO) or a priming dose (0.25 µg/50 µl sesame oil, *sc*) of E2 benzoate (Sigma-Aldrich, St. Louis, MO) in the morning. On the following day, oil or a high (1.5 µg) dose of E2 benzoate, which generates an induced LH surge, was administered (*Bosch et al., 2013*). Circulating levels of E2 were verified by the uterine weights (<25 mg for OVX and >95 mg for OVX E2-treated) at the time of euthanasia (*Bosch et al., 2013*).

## AAV delivery

Bilateral ARH injections of AAV1-Ef1a-DIO-ChR2:mCherry (RRID:Addgene_20297) or AAV1-Ef1a-DIO-ChR2:YFP (RRID:Addgene_20298) were performed on adult *Kiss1^Cre^* mice or *Pomc^Cre^* mice on a stereotaxic frame under isoflurane anesthesia. ARH injection coordinates were anteroposterior (AP): −1.10 mm, mediolateral (ML): ±0.30 mm, dorsoventral (DL): −5.80 mm (surface of brain $z$ = 0.0 mm); 400 nl of the AAV ($2.0 \times 10^{12}$ particles/ml) was injected (100 nl/min) into each position. Mice were given carprofen for analgesia and allowed to recover for at least 2 weeks before euthanasia.

## Visualized whole-cell patch recordings

Hypothalamic coronal brain slices (240 um) were made from female mice, using a Leica VT1000S vibratome, in ice-cold cutting solution bubbled with $O_2/CO_2$ (95%/5%). Slices were then transferred to a holding chamber with artificial cerebrospinal fluid bubbled with the same gas mix and allowed to recover for at least 1 hr. For recordings, slices were placed in a perfusion chamber and visualized with an Olympus BX51W1 using either differential infrared contrast or oblique illumination. Kiss1 neurons in the ARH or AVPV/PeN were targeted for electrophysiological recordings as done previously and described below (*Qiu et al., 2016*; *Stincic et al., 2021*). Final concentration was calculated based on the drug in the known volume of the bath. Perfused drugs (TTX, naloxone, and STX) were constantly circulated and given at least 10 min to reach maximal effect. Focal application of drugs (MTII and THIQ) was done with the pump stopped (10–20 min). 0.3–1 µl was directly added to the known volume of the bath to achieve the desired final concentration. This approach enables precise timing of recordings and reduces the risk of receptor desensitization.

## Solutions/drugs

Standard vibratome slicing, external, and internal recording solutions were utilized as previously described (*Qiu et al., 2016*; *Stincic et al., 2021*). TTX was purchased from Alomone Labs (Jerusalem, Israel), MTII and αMSH from Tocris (Minneapolis, MN). TEA, 4-AP, 17β-estradiol benzoate, and naloxone were purchased from Millipore-Sigma. STX was produced by AAPharmaSyn, LLC (Ann Arbor, MI) under contract.

## Electrophysiology data analysis

Electrophysiological data were analyzed using Clampfit 10/11 (Molecular Devices) and Prism 7/10 (Dotmatics). All values are expressed as mean ± SEM. Comparisons between two groups were made using unpaired Student's *t*-test or between multiple groups using an ANOVA (with post hoc comparisons) with p-values <0.05 considered significant. When variances differed significantly, the Mann–Whitney *U* test was used instead.

## Targeting of Kiss1 neurons for electrophysiological recordings

For non-optogenetic experiments, brain slices were taken from AAV-injected *Kiss1^Cre^* AAV or Kiss1xAi32 female mice. Ai32 mice (RRID:IMSR_JAX:024109, C57BL/6 background) carry the floxed

ChR2 (H134R)-EYFP gene in their Gt(ROSA)26Sor Locus (*Madisen et al., 2012*), allowing its expression in a Cre-dependent manner. Due to concerns of non-specific expression (*Qiu et al., 2021*), we previously validated this model using single cell RT-PCR and documented that *Kiss1* mRNA was detectable in 99% of individually harvested eYFP cells (*n* = 126). In addition, we have used both AAV-injected *Kiss1*^Cre AAV or Kiss1xAi32 female mice and found no differences in electrophysiological results (*Qiu et al., 2016*). We used AAV-injected POMC-Cre female mice and avoided small soma, high input resistance (>800 MΩ), ventrally located cells that are typically NPY/AgRP neurons. Instead, we targeted larger, more dorsomedial neurons in the ARH while avoiding fluorescent (POMC) cells. Unlike ARH Kiss1 neurons, POMC neurons do not make reciprocal projections, so uninfected POMC neurons do not display monosynaptic EPSCs in response to optogenetic stimulation. Next, we used a ramp IV protocol to probe for the presence of a persistent sodium current ($I_{NaP}$). While this current is more prevalent in the AVPV Kiss1 population, Kiss1 neurons are the only ARH neurons to display this electrophysiological 'fingerprint' of a pronounced $I_{NaP}$ paired with a high capacitance and low input resistance (*Zhang et al., 2015*). Finally, we also harvested the cytosol at the end of all recordings and measured the expression of Kiss1. Only cells that expressed a persistent sodium current and/or expressed *Kiss1* mRNA were included in the final analysis (*n* = 40 ARH and 6 AVPV).

## Acknowledgements

The authors would like to recognize the excellent technical expertise of Ms. Martha A Bosch (tissue preparation, immunohistochemical procedure, and single-cell RT-PCR of recorded neurons). In addition, we thank Dr. Rona Carroll for her assistance at Harvard Medical School and Mr. Cole Martinson, a student worker in the Ronnekleiv/Kelly laboratories, for his assistance with genotyping and care of mouse colonies at OHSU. The University of Virginia Center for Research in Reproduction Ligand Assay and Analysis Core is supported by the Eunice Kennedy Shriver NICHD/NIH (NCTRI) Grant P50-HD28934. This work was supported by PHS MPI grant DK68098 to MJK and OKR; HD090151, HD099084, and DK133760 to VMN, and The Charles A King Trust Postdoctoral Research Fellowship Award, The Lalor Foundation Postdoctoral Fellowship Award, the Women's Brain Initiative Fellowship Award, the ROSA SCORE pilot grant (supported by NIH Research Grant U54 AG062322 funded by The National Institute on Aging (NIA) and Office of Research on Women's Health (ORWH)) and the IBRO-ISN Research Fellowship Award to RT, and startup funds from Appalachian State University to TLS.

## Additional information

### Funding

| Funder | Grant reference number | Author |
| --- | --- | --- |
| National Institutes of Health | R01HD099084 | Victor M Navarro |
| National Institutes of Health | R01DK133760 | Victor M Navarro |
| National Institutes of Health | R01HD090151 | Victor M Navarro |
| National Institutes of Health | U54AG062322 | Victor M Navarro |
| Lalor Foundation | Postdoctoral Research Fellowship Award | Rajae Talbi |
| Charles A. King Trust | Postdoctoral Research Fellowship Award | Rajae Talbi |
| National Institutes of Health | R01DK068098 | Oline K Rønnekleiv Martin J Kelly |

The funders had no role in study design, data collection, and interpretation, or the decision to submit the work for publication.

## Author contributions
Rajae Talbi, Conceptualization, Data curation, Formal analysis, Validation, Investigation, Visualization, Methodology, Writing – original draft, Writing – review and editing; Todd L Stincic, Conceptualization, Data curation, Software, Formal analysis, Validation, Investigation, Methodology, Writing – original draft, Writing – review and editing; Kaitlin Ferrari, Choi Ji Hae, Karol Walec, Elizabeth Medve, Achi Gerutshang, Silvia Leon, Elizabeth A McCarthy, Investigation, Methodology; Oline K Rønnekleiv, Conceptualization, Resources, Data curation, Formal analysis, Investigation, Methodology, Writing – original draft, Writing – review and editing; Martin J Kelly, Conceptualization, Resources, Data curation, Formal analysis, Supervision, Funding acquisition, Validation, Investigation, Visualization, Methodology, Writing – original draft, Writing – review and editing; Victor M Navarro, Conceptualization, Resources, Data curation, Formal analysis, Supervision, Funding acquisition, Validation, Investigation, Visualization, Methodology, Writing – original draft, Project administration, Writing – review and editing

## Author ORCIDs
Rajae Talbi ⓘ https://orcid.org/0000-0001-7158-6246
Todd L Stincic ⓘ https://orcid.org/0000-0001-7504-2422
Oline K Rønnekleiv ⓘ https://orcid.org/0000-0003-1841-4386
Martin J Kelly ⓘ https://orcid.org/0000-0002-8633-2510
Victor M Navarro ⓘ https://orcid.org/0000-0002-5799-219X

## Ethics
The animal studies were approved by the Brigham and Women's Hospital Institutional Animal Care and Use Committee (IACUC) in the Center for Comparative Medicine. Adult wild-type (WT) C57/BL6 female mice were group housed under constant conditions of temperature (22–24°C) and light (12:12-hr light:dark cycle), fed with standard mouse chow and ad libitum access to tap water. All animal procedures described in the electrophysiology studies were performed in accordance with institutional guidelines based on National Institutes of Health standards and approved by the Institutional Animal Care and Use Committees at Oregon Health and Science University and Appalachian State University.

Reviewer #2 (Public review): https://doi.org/10.7554/eLife.100722.4.sa1
Author response https://doi.org/10.7554/eLife.100722.4.sa2

# Additional files

## Supplementary files
Source data 1. Electrophysiology raw data.

MDAR checklist

Source code 1. MATLAB code used to analyze the luteinizing hormone (LH) pulses data (in *Figures 2 and 3*).

## Data availability
All data generated or analyzed during this study are included in the manuscript and supporting files.

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
