## [Editor Report · eLife Assessment]

The study presents **compelling** evidence that the melanocortin system originating in the arcuate nucleus of the hypothalamus plays a crucial role in puberty onset, representing a significant advance in our understanding of reproductive biology. The research employs innovative approaches and benefits from the combined expertise of two respected laboratories, enhancing the robustness of the findings. Given the potential impact on human health and the strength of the evidence presented, this **fundamental** work will likely influence the field substantially and may inform future clinical applications.

---

## [Referee Report · Reviewer #2 (Public review)]

Summary:

I found this an interesting manuscript describing a study investigating the role of MC4R signalling on kisspeptin neurons. The initial question is a good one. Infertility associated with MC4 mutations in humans has typically been ascribed to the consequent obesity and impaired metabolic regulation. Whether there is a direct role for MC4 in regulating the HPG axis has not been thoroughly examined. Here, the researchers have put together an elegant combination of targeted loss of function and gain of function in vivo experiments, specifically targeting MC4 expression in kisspeptin neurons. This excellent experimental design should provide compelling evidence for whether melanocortin signalling has a direct role in arcuate kisspeptin neurons to support normal reproductive function. There were definite effects on reproductive function (irregular estrous cycle, reduced magnitude of LH surge induced by exogenous estradiol). However, the magnitude of these responses and the overall effect on fertility were relatively minor. The mice lacking MC4R in kisspeptin neurons remained fertile despite these irregularities. The second part of the manuscript describes a series of electrophysiological studies evaluating the pharmacological effects of melanocortin signalling in kisspeptin cells in ex-vivo brain slides. These studies characterised interesting differential actions of melanocortins in two different populations of kisspeptin neurons. Collectively, I think the study provides novel insights into how direct actions of melanocortin signalling, via the MC4 receptor in kisspeptin neurons, contribute to the metabolic regulation of the reproductive system. Importantly, however, it is clear that other mechanisms are also at play.

Strengths:

The loss of function/gain of function experiments provide a conceptually simple but hugely informative experimental design. This is the key strength of the current paper - especially the knock-in study that showed improved reproductive function even in the presence of ongoing obesity. This is a very convincing result that documents that reproductive deficits in MC4R knockout animals (and humans with deleterious variants of the MC4R gene) can be ascribed to impaired signalling in the hypothalamic kisspeptin neurons and not necessarily simply caused as a consequence of obesity. As concluded by the authors: "reproductive impairments observed in MC4R deficient mice, which replicate many of the conditions described in humans, are largely mediated by the direct action of melanocortins via MC4R on Kiss1 neurons and not to their obese phenotype." This is important, as it might change the way such fertility problems are treated.

Limitation:

The mechanistic studies evaluating melanocortin signalling in kisppetin neurons were all completed in ovariectomized animals (with and without exogenous hormones). This reductionist approach allowed a focus on the direct actions of estradiol to regulate responses but missed an opportunity to evaluate how cyclical changes in hormones might impact the system. Such cyclical changes are fundamental to how these neurons function in vivo and may dynamically alter the way they respond to hormones and neuropeptides. However, the inclusion of gonad-intact animals would have significantly increased the complexity of experiments and can reasonably be considered outside of the scope of the present study.

---

## [Author Response]

The following is the authors’ response to the previous reviews

**Public Reviews:**

**Reviewer #1 (Public review):**
Summary:The authors investigate the role of the melanocortin system in puberty onset. They conclude that POMC neurons within the arcuate nucleus of the hypothalamus provide important but differing input to kisspeptin neurons in the arcuate or rostral hypothalamus.Strengths:Innovative and novelTechnically soundWell-designedThoroughWeaknesses:There were no major weaknesses identified.
**Reviewer #2 (Public review):**
Summary:This interesting manuscript describes a study investigating the role of MC4R signalling on kisspeptin neurons. The initial question is a good one. Infertility associated with MC4 mutations in humans has typically been ascribed to the consequent obesity and impaired metabolic regulation. Whether there is a direct role for MC4 in regulating the HPG axis has not been thoroughly examined. Here, the researchers have assembled an elegant combination of targetted loss of function and gain of function in vivo experiments, specifically targetting MC4 expression in kisspeptin neurons. This excellent experimental design should provide compelling evidence for whether melanocortin signalling dirently affects arcuate kisspeptin neurons to support normal reproductive function. There were definite effects on reproductive function (irregular estrous cycle, reduced magnitude of LH surge induced by exogenous estradiol). However, the magnitude of these responses and the overall effect on fertility were relatively minor. The mice lacking MC4R in kisspeptin neurons remained fertile despite these irregularities. The second part of the manuscript describes a series of electrophysiological studies evaluating the pharmacological effects of melanocortin signalling in kisspeptin cells in ex-vivo brain slides. These studies characterised interesting differential actions of melanocortins in two different populations of kisspeptin neurons. Collectively, the study provides some novel insights into how direct actions of melanocortin signalling via the MC4 receptor in kisspeptin neurons contribute to the metabolic regulation of the reproductive system. Importantly, however, it is clear that other mechanisms are also at play.Strengths:The loss of function/gain of function experiments provides a conceptually simple but hugely informative experimental design. This is the key strength of the current paper - especially the knock-in study that showed improved reproductive function even in the presence of ongoing obesity. This is a very convincing result that documents that reproductive deficits in MC4R knockout animals (and humans with deleterious MC4R gene variants) can be ascribed to impaired signalling in the hypothalamic kisspeptin neurons and not necessarily caused as a consequence of obesity. As concluded by the authors: "reproductive impairments observed in MC4R deficient mice, which replicate many of the conditions described in humans, are largely mediated by the direct action of melanocortins via MC4R on Kiss1 neurons and not to their obese phenotype." This is important, as it might change how such fertility problems are treated.I would like to see the validation experiments for the genetic manipulation studies given greater prominence in the manuscript because they are critical to interpretation. Presently, only single unquantified images are shown, and a much more comprehensive analysis should be provided.Weaknesses:(1) Given that mice lacking MC4R in kisspeptin neurons remained fertile despite some reproductive irregularities, this can be described as a contributing pathway, but other mechanisms must also be involved in conveying metabolic information to the reproductive system. This is now appropriately covered in the discussion.(2) The mechanistic studies evaluating melanocortin signalling in kisspeptin neurons were all completed in ovariectomised animals (with and without exogenous hormones) that do not experience cyclical hormone changes. Such cyclical changes are fundamental to how these neurons function in vivo and may dynamically alter how they respond to hormones and neuropeptides. Eliminating this variable makes interpretation difficult, but the authors have justified this as a reductionist approach to evaluate estradiol actions specifically. However, this does not reflect the actual complexity of reproductive function.For example, the authors focus on a reduced LH response to exogenous estradiol in ovariectomised mice as evidence that there might be a sub-optimal preovulatory LH surge. However, the preovulatory LH sure (in intact animals) was not measured.They have not assessed why some follicles ovulated, but most did not. They have focused on the possibility that the ovulation signal (LH surge) was insufficient rather than asking why some follicles responded and others did not. This suggests some issue with follicular development, likely due to changes in gonadotropin secretion during the cycle and not simply due to an insufficient LH surge.
**Reviewer #3 (Public review):**
The manuscript by Talbi R et al. generated transgenic mice to assess the reproduction function of MC4R in Kiss1 neurons in vivo and used electrophysiology to test how MC4R activation regulated Kiss1 neuronal firing in ARH and AVPV/PeN. This timely study is highly significant in neuroendocrinology research for the following reasons.(1) The authors' findings are significant in the field of reproductive research. Despite the known presence of MC4R signaling in Kiss1 neurons, the exact mechanisms of how MC4R signaling regulates different Kiss1 neuronal populations in the context of sex hormone fluctuations are not entirely understood. The authors reported that knocking out Mc4r from Kiss1 neurons replicates the reproductive impairment of MC4RKO mice, and Mc4r expression in Kiss1 neurons in the MC4R null background partially restored the reproductive impairment. MC4R activation excites Kiss1 ARH neurons and inhibits Kiss1 AVPV/PeN neurons (except for elevated estradiol).(2) Reproduction dysfunction is one of obesity comorbidities. MC4R loss-of-function mutations cause obesity phenotype and impaired reproduction. However, it is hard to determine the causality. The authors carefully measured the body weight of the different mouse models (Figure 1C, Figure 2A, Figure 3B). For example, the Kiss1-MC4RKO females showed no body weight difference at puberty onset. This clearly demonstrated the direct function of MC4R signaling in reproduction but was not a consequence of excessive adiposity.(3) Gene expression findings in the "KNDy" system align with the reproduction phenotype.(4) The electrophysiology results reported in this manuscript are innovative and provide more details of MC4R activation and Kiss1 neuronal activation.Overall, the authors have presented sufficient background in a clear, logical, and organized structure, clearly stated the key question to be addressed, used the appropriate methodology, produced significant and innovative main findings, and made a justified conclusion.Comments on revisions:The authors have addressed my comments.
**Recommendations for the authors:**
The reviewers noted that they received comments in response to their concerns, and some improvements have been made to the manuscript. However, as described below, in some cases, a rebuttal was provided, but changes were not made to the manuscript. It is suggested that these issues be addressed to improve the quality of the manuscript.

We thank the reviewers and editor for the assessment of the manuscript and recommendations for its improvement. We have addressed the remaining comments from reviewer #2 below, and hope that they find our revisions satisfactory.

**Reviewer #2 (Recommendations for the authors):**
The manuscript convincingly shows that MC4R in kisspeptin-producing cells can influence reproductive function. This suggests that fertility problems associated with melanocortin mutations are likely due to direct effects on the reproductive systems rather than simply being side effects of the resultant obesity.

We are pleased that this reviewer finds the data convincing and thank them for the careful review of the manuscript, which has helped to improve its published version.

The authors have responded to the reviewer's comments and made several improvements to the manuscript.The authors are correct in pointing out that the POMC-Cre animals should be fine for studies involving the administration of AAVs to adult animals. I have misinterpreted how these mice were being used, and this concern is fully addressed.Unfortunately, in some cases, the authors rebutted the reviewer's comments but did not change the manuscript. I suggest addressing several issues in the manuscript (after all, it is not the reviewer's opinion that counts; this process is about improving the manuscript).(1) Validation of the KO is insufficiently reported. From the methods, it appears that this was done thoroughly, but currently, only a single image of the arcuate nucleus is shown, and no image of the AVPV is shown. There is no quantitative information provided. The authors can keep these data as supplementary material, but they should be comprehensive and convincing, as so much depends on the degree of knockout in this model. One cannot assume complete KO based simply on the relevant genetics, as there are examples in this system where different Cre lines produce different outcomes with various floxed genes in the two major populations of kisspeptin neurons. This figure should show the quantitation of the RNAscope analysis from each of the two regions regarding the percentage of kisspeptin cells showing expression of MC4R mRNA. In addition, the lack of MC4 labelling in the arcuate nucleus, outside of kisspeptin neurons, is a concern. One would expect to see AgRP or POMC cells at this level, but are they still showing expression of MC4? A single image is insufficient to be convinced of the model's efficacy.

We appreciate the reviewer’s concerns regarding the validation of the MC4RKO model. Below, we provide clarification and additional justification for our approach.

(1) Quantification of MC4R in the Arcuate Nucleus (ARC): As noted by the reviewer, we were unable to detect sufficient MC4R signal in the ARC of KO mice to perform meaningful quantification. This is consistent with the expected outcome of a successful MC4R deletion. Given the low endogenous expression levels of MC4R in this region, even in control animals, and the technical limitations of RNAscope in detecting very low-abundance transcripts, especially for receptors, the absence of MC4R signal in the ARC of KO mice strongly supports effective deletion. Moreover, the MC4R loxP mouse has been published and validated by many labs including Brad Lowell’s lab who’s done extensive work using these mice for selective deletion of Mc4r from various neuronal populations such as Sim1 and Vglut2 neurons (Shah et al., 2014, de Souza Cordeiro et al., 2020). To further strengthen our validation, we provide additional images from another animal (Fig_S1) to illustrate the consistency of the MC4R KO in the ARC. These will be included as supplementary material, as suggested.Regarding AgRP and POMC neurons, MC4R is not highly expressed in these neurons (as per previous literature, e.g., Garfield et al., Nat Neurosci. 2015; Padilla SL et al, Endocrinology 2012; Henry et al, Nature, 2015). Instead, MC4R is predominantly found in downstream neurons in the paraventricular nucleus (PVN) and other hypothalamic regions (which is intact in our KO mice as shown in our validation figure). Thus, the absence of MC4R labeling in AgRP or POMC cells in our images aligns with known expression patterns and does not contradict the validity of our model.

(2) MC4R Expression in the AVPV and OVX Effect on Kiss1 Expression: We acknowledge the reviewer’s request for MC4R expression analysis in the anteroventral periventricular nucleus (AVPV). However, due to the timing of tissue collection after ovariectomy (OVX), Kiss1 expression in the AVPV is significantly suppressed, making it technically unfeasible to perform co-staining of MC4R with Kiss1 in this region. This is a well-documented effect of estrogen depletion following OVX (Smith et al., 2005; Lehman et al., 2010). While we acknowledge that an ideal validation would include AVPV co-labeling, the experimental constraints related to OVX preclude this analysis in our dataset.

Given these considerations and validations, we are confident that the KO is effective and specific.

(2) Line 88: "... however, conflicting reports exist". Expand on this sentence to describe what these conflicting reports show. The authors responded to my comment but made no changes to the introduction. As a reader, I dislike being told there are conflicting reports, but then I have to go and look up the reference to see what that actual point of conflict is.

By conflicting reports we meant that other studies have shown no association between MC4R and reproductive disorders, this has now been included in the revised manuscript (Line 89).

(3) Could the authors explain how a decrease in AgRP would be interpreted as a "decrease in hypothalamic melanocortin tone" in line 142 and line 364? These overly simplistic interpretations of qPCR data detract from the overall quality of the paper.

The reference to a decrease in melanocortin tone referred to the decrease in the expression of melanocortin receptor signaling, this has been clarified in the revised manuscript (lines 142 and 360).

(4) Please show the individual cycle patterns for all animals, as in Figure 2B. This can be a supplemental figure, but the current bar charts are not informative.

We respectfully disagree that the bar charts are not informative as they include the critical statistical analysis. We have now included all individual estrous cycle data in new separate supplemental figure (Sup. Figure 3). Therefore, we have excluded the representative cycles from the main figures as they are now in the new Supplemental. We have changed the orders of the figures in the text accordingly.

(5) In their rebuttal, the authors state: "Mice lack true follicular and luteal phases, and therefore, it is impossible to separate estrogen-mediated changes from progesterone-mediated changes (e.g., in a proestrous female). Therefore, we use an ovariectomized female model in which we can generate an LH surge with an E2-replacement regimen [1]. This model enables us to focus on estrogen effects, exclude progesterone effects, and minimize variability. Inclusion of cycling females would make interpretation much more difficult." I disagree, but the authors can take this position if they wish. However, they should not report the responses to exogenous estradiol in an ovariectomised mouse as a "preovulatory LH surge" (line 380). An ovariectomised mouse cannot ovulate, and the estrogen-induced LH surge is significantly different in magnitude and timing from the endogenous preovulatory LH surge (likely due to the actions of progesterone). One goal of these studies is to understand why the ovulation rate appears to be low in the MC4-KO animals. Hence, evaluating whether the preovulatory LH surge is typical is important. This has not been done. The authors have shown that the response to exogenous estradiol is sub-normal. Such an effect might lead to a reduced preovulatory LH surge, but this has not been measured.

We appreciate this reviewer’s concern about the nature of the preovulatory LH surge. We have clarified this in the revised manuscript and described it as “an induced LH surge” throughout the text (Lines 163, 533, 6560).

(6) I believe that the ovulation process should be considered "all or none," and I do not quite understand the rebuttal discussion. The authors describe that "numerous follicles mature at the same time....". That is not disputed. My point was that each mature follicle will receive the identical endocrine ovulatory signal (correct? Or do the authors believe something different?). If it were sufficient for one follicle to ovulate, then all of those mature follicles (the number of which will be variable between animals and between cycles) would be expected to undergo ovulation. The fact that they do not raise several possibilities. One that the authors favor is that an insufficient ovulatory signal might approach a threshold where some follicles ovulate and others do not. This possibility is supported by the apparent increase in cystic follicles, which might be preovulatory follicles that did not complete the ovulation process. Such variation might be stochastic, within normal variation for sensitivity to LH. However, it is also possible that the follicles have not matured at the same rate, perhaps influenced by abnormal secretion of LH or FSH during earlier phases of the cycle, and hence are not in the appropriate condition to respond to the ovulation signal when it arrives. Some may even have matured prematurely due to the elevated gonadotropins reported in this study. Given the data and the partial fertility, the most likely explanation is that the genetic manipulation has resulted in fewer follicles being available for ovulation due to changes in follicular development rather than a deficit of the ovulation signal, although the latter mechanism might also contribute. A third possibility is that genetic manipulation has directly affected the ovary. The authors did not answer whether Kiss1 and MC4 are co-expressed in the ovary. I think the authors might want to rule this out by showing no change in MC4R expression in the ovary.

We thank the reviewer for this thoughtful comment and agree that these are possible outcomes. We have now acknowledged them in the Discussion.

To answer the reviewer’s question, we have not investigated the co-expression of Kiss1 and Mc4r in the ovary. While MC4R has indeed been documented in the ovary (Chen et al. Reproduction, 2017), the changes in gonadotropin release and supporting in vitro data included in this manuscript clearly document a central effect, however, an additional effect at the level of the ovary cannot be completely ruled out. This has now been added to the discussion (Line 378-387).

(7) Lines 390, 454 " impaired LH pulse" What was the evidence for impaired LH pulse (see figure 2D)?

Thank you for pointing this out. This comment referred to augmented LH release. This has been corrected in the revised manuscript (Line 394).

The paper's strengths remain, as outlined in my original review. The authors have addressed what I perceived to be weaknesses, predominantly by changing the tone of discussion and interpretation of the data. This is appropriate. I consider the focus on the LH surge as the primary mechanism too narrow, and the authors should be considering how other changes during the cycle might influence ovarian function.

We sincerely appreciate the reviewer’s thoughtful evaluation of our manuscript and their constructive feedback. We are pleased that our revisions have addressed the perceived weaknesses and that the adjustments to the discussion and interpretation were deemed appropriate.

We acknowledge the reviewer’s perspective on broadening the discussion beyond the LH surge to consider additional cycle-dependent influences on ovarian function. While our current study focuses on this specific mechanism, we recognize that ovarian function is influenced by multiple physiological changes throughout the cycle. We have refined our discussion to reflect this broader context and appreciate the suggestion to consider these additional factors in future studies.

We have addressed all of the reviewer’s comments to the best of our ability and hope they find the revised manuscript satisfactory.